# UCLALES-SALSA v1.0: a large-eddy model with interactive sectional microphysics for aerosol, clouds and precipitation

Juha Tonttila[1,2], Zubair Maalick[4], Tomi Raatikainen[3], Harri Kokkola[2], Thomas Kühn[4], and Sami Romakkaniemi[2]

[1]Karlsruhe Institute of Technology, Karlsruhe, Germany
[2]Finnish Meteorological Institute, Atmospheric Research Centre of Eastern Finland, P.O. Box 1627, 70211 Kuopio, Finland
[3]Finnish Meteorological Institute, P.O. Box 503, 00101 Helsinki, Finland
[4]Department of Applied Physics, University of Eastern Finland, P.O. Box 1627, 70211 Kuopio, Finland

*Correspondence to:* Juha Tonttila (juha.tonttila@fmi.fi)

**Abstract.** Challenges in understanding the aerosol-cloud interactions and their impacts on global climate highlight the need for improved knowledge of the underlying physical processes and feedbacks as well as their interactions with cloud and boundary layer dynamics. To pursue this goal, increasingly sophisticated cloud-scale models are needed to complement the limited supply of observations of the interactions between aerosols and clouds. For this purpose, a new large-eddy simulation (LES) model, coupled with an interactive sectional description for aerosols and clouds, is introduced. The model, UCLALES-SALSA, builds and extends upon a well characterized LES model (UCLALES) and microphysical model components (SALSA). Novel strategies for the aerosol, cloud and precipitation bin discretization are presented. These enable tracking the effects of cloud processing and wet scavenging on the aerosol size distribution as accurately as possible while keeping the computational cost of the model as low as possible. The model is tested with two different simulation setups: a marine stratocumulus case in the DYCOMS-II campaign and another case focusing on the formation and evolution of a nocturnal radiation fog. It is shown that, in both cases, the size-resolved interactions between aerosols and clouds have a critical influence on the dynamics of the boundary layer. The results demonstrate the importance of accurately representing the wet scavenging of aerosol in the model. Specifically, in a case with marine stratocumulus, precipitation and the subsequent removal of cloud activating particles lead to thinning of the cloud deck and the formation of a decoupled boundary layer structure. In radiation fogs, the growth and sedimentation of droplets strongly affect their radiative properties, which in turn drive new droplet formation. The size resolved diagnostics provided by the model enable investigations of these issues with high detail. It is also shown that the results remain consistent with UCLALES (without SALSA) in cases, where the dominating physical processes remain well represented by both models.

## 1   Introduction

Large eddy simulations (LES) have been used to study the properties of clouds and the boundary layer for a few decades (e.g. Deardorff, 1974, 1980; Moeng, 1984; Stevens et al., 2005). These models solve the low-pass filtered Navier-Stokes equations, i.e. the large energy-containing turbulent eddies are resolved, while the smallest length scales and energy dissipation are

parameterized typically using closures based on the Smagorinsky model. This approach provides an attractive compromise between accuracy and computational cost, which is why LES models have become popular in studies of the properties of boundary layers and clouds.

The typical grid resolution used in LES models (on the order of tens of meters) enables a detailed representation of cloud structure and dynamics. However, the treatment of cloud microphysics is subject to high variability in terms of the level of detail and computational cost (Khain et al., 2015). The types of microphysical schemes and their implementation to LES models range from simple one or two moment bulk schemes, where droplet mass is predicted typically through saturation adjustment, with either prescribed or varying droplet number concentrations (Khairoutdinov and Kogan, 2000; Golaz et al., 2005; Seifert and Beheng, 2001, 2006; Stevens et al., 2005; Savre et al., 2014), to more elaborate ones with modal or sectional representations for the droplet size distributions (Feingold et al., 1996; Feingold and Kreidenweis, 2002; Saleeby et al., 2015) and Lagrangian particle based methods (Shima et al., 2009). In addition, there has been an increasing trend towards including representations for aerosol particles in these models as well (Feingold and Kreidenweis, 2002; Kazil et al., 2011; Maalick et al., 2016). However, extensive simulations with more detailed and explicitly interactive aerosol-cloud schemes are as of yet relatively sparse, mostly due to their high computational cost. Nevertheless, some examples of such developments include the works of (Andrejczuk et al., 2010; Ovchinnikov and Easter, 2010; Kazil et al., 2011; Lebo and Seinfeld, 2011; Vié et al., 2016).

The need for such models is well recognized due to the significant challenges in climate modelling imposed by aerosols and clouds (Boucher et al., 2013), where detailed LES model simulations comprise an essential resource for parameterization develoment. In particular, formation of drizzle and wet scavenging of aerosol and the associated feedback processes are potentially very important for the dynamics and circulation structures of marine stratocumulus clouds (Wang et al., 2010; Wood and Bretherton, 2004; Wood et al., 2012; Terai et al., 2014). Correctly capturing the interactions between aerosol-cloud microphysics and cloud dynamics requires highly detailed microphysical schemes. Moreover, scavenging processes, depending on particle composition and size, are overall rather poorly understood and therefore poorly represented in general circulation models (Boucher et al., 2013; Croft et al., 2010). Yet, wet scavenging of aerosol may crucially affect e.g. the transport of black carbon aerosol from polluted environments to the polar areas (Garrett et al., 2011), where it has the potential to significantly affect the future change in arctic temperatures. The main motivation for the LES model development presented in the current paper is indeed to provide a new tool for a better understanding of the above mentioned climate-relevant processes, so that they can eventually be more robustly represented in global models.

Besides cloud processes, another set of topics under research by the LES community is related to the formation and evolution of fogs and the effects of aerosols therein. During the last decades, a clear decrease has been observed in fog occurrence throughout Central Europe (Vautard et al., 2009; Giulianelli et al., 2014). This has occurred together with improved air quality due to a decreasing trend in sulfur emissions, especially in the case of dense fogs, but this far, a quantitative connection has not been established (van Oldenborgh et al., 2010). Although in many ways driven by the same principles as clouds, fogs also feature many unique aspects considering their evolution. (Nakanishi, 2000; Gultepe et al., 2007). For example, while cloud droplets are mainly formed at the height of the peak saturation ratio at cloud base, in radiation fogs, one of the most common fog types, the droplet formation is primarily driven by radiative cooling at the top of the developing fog layer or by

high supersaturation inside the fog induced by turbulence. Thus there are also marked differences related to the dynamics of the fog layer and it's life cycle as compared to clouds (Porson et al., 2011). Fog properties and their occurrence are strongly affected by aerosol properties and anthropogenic emissions (Bott, 1991; Kokkola et al., 2003; Stolaki et al., 2015; Maalick et al., 2016), although many of the details of these interactions remain poorly understood. Improved knowledge can be pursued through increasingly sophisticated microphysical schemes embedded in LES models. Thus, a case comprising a radiation fog event serves as a well-justified testbed for the model presented in this paper.

Here, an innovative approach is proposed to treat the microphysical interactions between aerosols and clouds as well as their impacts on boundary layer dynamics within a high-resolution LES model while keeping the model computationally feasible for long simulation times (few days) and large model domains (tens of kilometers). We build and extend upon a state-of-the-art LES model and a sectional microphysical model (Stevens et al., 2005; Kokkola et al., 2008) to create a cloud-resolving framework, where the size distributions of aerosol, clouds and precipitation are all described with a detailed sectional approach. In particular, the model introduced in this work accurately preserves the characteristics of the aerosol size distribution both in- and outside of clouds, making it ideal for studying the impact of removal processes, cloud processing and evaporation on the particle size distribution, as well as the associated feedbacks on cloud properties, precipitation formation and boundary layer dynamics. The model is evaluated by experimenting on two very different cases: one comprising marine stratocumulus clouds based on the DYCOMS-II dataset (Stevens et al., 2003), and another focusing on a radiation fog event based on the findings of (Porson et al., 2011; Price, 2011). The results are compared with earlier studies and models with a simple bulk microphysics scheme, and similarities and differences are analyzed and explained in detail.

The new model is described in detail in Section 2 while case descriptions and results for the marine stratocumulus and fog cases are documented in Sections 3 and 4, respectively. Discussion of the model performance and conclusions drawn from the results are reported in Section 5.

## 2 Model description

### 2.1 The extended SALSA module

The Sectional Aerosol module for Large Scale Applications (SALSA; Kokkola et al. (2008)) is used as the basis for developing a unified sectional microphysical model for aerosols, clouds and precipitation. The SALSA module, previously employed in the ECHAM (Stevens et al., 2013) climate model family, discretizes the aerosol size distribution into 10 size bins according to the dry particle diameter (Bergman et al., 2012) as shown in Figure 1. The predicted variables for each bin are the aerosol number and compound masses as well as the mass of condensed water, which can be used to determine the bin mean wet particle size. The total diameter range covered by the bins (from $3\,\mathrm{nm}$ to $10\,\mu\mathrm{m}$ by default) is divided into subranges, 1a and 2a. This division into subranges aims at minimizing the number of tracer variables. This is achieved by including only those chemical compounds that are significantly abundant in each subrange. Subrange 1a covers the three smallest bins (up to $50\,\mathrm{nm}$) and the particles are assumed to be internally mixed, being composed of sulfate and organic carbon, which contribute to the growth of newly formed particles. Subrange 2a includes particles larger than $50\,\mathrm{nm}$ whose composition may comprise all the

chemical compounds in the model. The module can be configured to include 7 additional bins (designated 2b) parallel to the bin regime 2a (i.e. same bin diameters), which allow the description of externally mixed particle populations. In a typical example, soluble compounds would be emitted to 2a and insoluble compounds to 2b. The spacing of the size bins is set logarithmically equidistant within each of the subranges. Further details about the bin discretization can also be found in Laakso et al. (2016).

With these settings, the spectral resolution is quite coarse, but does provide a good compromise between computational cost and model performance. Note however, that the numbers given here represent the default settings - the number of bins can be set to be larger, if necessary.

The SALSA module includes detailed methods for solving the key microphysical processes which are called sequentially. Coagulation is modelled based on the equations in Jacobson (2005). For particle number this is given as

$$
\quad n_{i,t} = \frac{n_{i,t-1}}{1 + \Delta t \sum\limits_{j=i+1}^{J} K_{i,j} n_{j,t-1} + \frac{1}{2} \Delta t K_{i,i} n_{i,t-1}}. \tag{1}
$$

Similarly, for volume concentration

$$
v_{i,t} = \frac{v_{i,t-1} \Delta t \sum\limits_{j=1}^{i-1} K_{j,i} v_{j,t} n_{i,t-h}}{1 + \Delta t \sum\limits_{j=i+1}^{J} K_{i,j} n_{j,t-1}}. \tag{2}
$$

In the above equations, $K_{i,j}$ is the total coagulation kernel for the colliding particles in bins $i$ and $j$, $n_{i,t}$ is the particle number concentration in bin $i$ at timestep $t$ ($t-1$ refers to the previous timestep), $v_{i,t}$ is the volume concentration, $\Delta t$ the length of the 15 timestep and $J$ is the total number of particle bins. Please note that the bin indices $1 \ldots J$ should be interpreted to cover all the bins of all particle categories in the model (aerosol, clouds, precipitation) sorted by increasing particle size.

For coagulation kernels with aerosol particles, we assume brownian coagulation, whose kernel in the continuum regime is given as

$$
K_{i,j}^B = 2\pi(d_i + d_j)(D_{p,i} + D_{p,j}), \tag{3}
$$

where $d_i$ and $d_j$ are the diameters of the colliding particles and $D_{p,i}$ and $D_{p,j}$ their corresponding diffusion coefficients. For the transition regime, the formula by Fuchs (1964) is used. For larger particles, i.e. cloud droplets and precipitation, the convective enhancement of Brownian coagulation and gravitational collection are also included, and given as

$$
K_{i,j}^{BC} = \begin{cases} K_{i,j}^B 0.45 Re_j^{1/3} Sc_i^{1/3} & Re_j \leq 1, d_j \geq d_i \\ K_{i,j}^B 0.45 Re_j^{1/2} Sc_i^{1/3} & Re_j > 1, d_j \geq d_i \end{cases} \tag{4}
$$

and

$$
\quad K_{i,j}^{GC} = E_{i,j}^c \pi (\frac{d_i + d_j}{2})^2 |V_{f,i} - V_{f,j}|, \tag{5}
$$

respectively (Jacobson, 2005). In the above equations, $Sc_i$ is the particle Schmidt number, $Re_j$ is the Reynolds number, $E_{i,j}^c$ is the collection efficiency and $V_{f,i}$ is the particle fall speed. The latter is parameterized as

$$V_{f,i} = \begin{cases} \frac{d_i^2(\rho_p - \rho_a)g\beta}{18\gamma} & d_i < 40\ \mu m \\ 2 \times 10^3 d_i (\frac{\rho_{a,ref}}{\rho_a})^2 & d_i \geq 40\ \mu m \end{cases} \tag{6}$$

where $\rho_p$ is the particle/droplet density and $\rho_a$ is the air density. $\rho_{a,ref}$ is a reference air density (given at the STP conditions, 273.15 K and 1000 hPa), $g$ is the gravitational acceleration, $\gamma$ is the dynamic viscosity of air and $\beta$ is the Cunningham slip correction factor. The total coagulation kernels in Equations (1) and (2) are obtained as the sum $K_{i,j} = K_{i,j}^B + K_{i,j}^{BC} + K_{i,j}^{GC}$. All the coagulation kernels are currently updated each timestep. However, this is computationally inefficient and the use of lookup tables with billinear interpolation in particle size is planned.

Condensation of water vapour and aerosol precursors gases (currently sulfuric acid and organics) is based on the Analytical Predictor of Condensation (APC) scheme by Jacobson (2005). The scheme first calculates the new vapour mole concentration as

$$C_t = \frac{C_{t-1} + \Delta t \sum_{i=1}^{J} k_{i,t-1} S_{i,t-1} C_{s,i,t-1}}{1 + \Delta t \sum_{i=1}^{J} k_{i,t-1}}, \tag{7}$$

where $k_{i,t-1}$ is the mass transfer coefficient in size bin $i$ based on the current timestep, $J$ is the total number of bins (including all particle categories), $S_{i,t-1}$ is the equilibrium supersaturation and $C_{s,i,t-1}$ is the saturation mole concentration over a flat surface. The new particle mole concentrations for each condensing vapour are then given in a semi-implicit form

$$c_{i,t} = c_{i,t-1} + \Delta t k_{i,t-1}(C_t - S_{i,t-1} C_{s,i,t-1}) \tag{8}$$

While the APC scheme is mass preserving and numerically stable, condensation and evaporation of water vapour on small droplets and especially small aerosol particles requires a very short timestep ($\ll 1$ s) to avoid non-oscillatory solutions. Since this goes beyond the practical range for the applications in this paper, where in general we aim towards a timestep around 1 s, two sets of measures are taken. First, for small aerosol particles with ambient relative humidity (RH) below 98 % the wet size of aerosol particles is determined as an equilibrium solution based on the molalities of different particle species (Stokes and Robinson, 1996; Kokkola et al., 2008), and the APC equations are solved only above 98 % RH. Second, a simple substepping method is applied with Equations (7) and (8), where the substep length for cloud droplets is user defined and currently set as $\Delta t_c = 0.1$ s. For non-activated aerosol above the 98 % threshold for RH, even further timesplitting was found necessary and the timescale is $\Delta t_a = t_c/10$. The equilibrium saturation ratio is updated for each substepping cycle due to changing droplet/particle size (temperature is kept constant).

Although not used in the context of this paper, new particle formation by sulfuric acid is included in the model. There, the activation-type nucleation is formulated according to Riipinen et al. (2007) and the formation rate of 3 nm particles is calculated according to Lehtinen et al. (2007).

### 2.1.1 Cloud droplets

In the new extended SALSA, cloud droplets are treated with a sectional description as well (Figure 1). Strictly speaking, to reproduce the evolution of the aerosol size distribution through cloud processing and wet scavenging accurately, which is the goal of this work, a two dimensional dry/wet diameter bin system would be required. This is because cloud activation depends essentially on the dry aerosol size distribution, while collision processes and deposition rates depend strongly on the wet particle size. Although such two-dimensional frameworks have been developed (e.g. Lebo and Seinfeld, 2011), the approach is computationally highly demanding for large-eddy modelling applications spanning timescales of days while covering relatively large domains with high spatial resolution, all of which are pursued here. As a compromise between accuracy and computational cost, a unique strategy is proposed, where cloud droplets are described based on the dry size of the activated aerosol (i.e. cloud condensation nuclei, CCN) with the same prognostic bin quantities as for the aerosol bins. The particle diameters at the bin edges for the cloud droplet and non-activated aerosol regimes are set identical within their common size range (specifically, the 2a/b-bins as a default setting) as shown in Figure 1. Therefore, each cloud droplet bin is accompanied by a parallel aerosol bin. This way, the shape of the aerosol size distribution and the number concentration are preserved upon cloud droplet activation as well as upon droplet evaporation though subject to the typical uncertainties inherent to the sectional approach (Khain et al., 2015). While the CCN dry diameter is known accurately (to the extent allowed by the spectral resolution of the size sections), the wet size of the cloud droplets, determined by Equations (1-8), represents a mean over each CCN size class.

Two methods are available for simulating the formation of cloud droplets in the extended SALSA. One is the parameterization by Abdul-Razzak and Ghan (2002), which takes as an input the aerosol properties and updraft velocity (along with atmospheric thermodynamic properties) to determine the maximum supersaturation in a parcel of air and thus the critical particle diameter for activation. Another is based on resolving the wet aerosol particle diameter: once the wet diameter of a particle exceeds the critical diameter corresponding to the resolved supersaturation from the host model, the particle is activated. Since the condensation of water vapour is solved dynamically for high RH, it is preferable to use the latter approach instead of the parameterized one for consistency in terms of the peak supersaturation and it is the approach used in the experiments of this work. This allows droplet activation also in other parts of the cloud apart from the cloud base, e.g. due to radiative cooling effects at the cloud top or supersaturation caused by mixing of airmasses. However, if the vertical resolution of the host model is coarse (several tens of meters and above) it becomes necessary to use the parameterized method. With coarse resolutions the supersaturation peak at the cloud base may be underestimated due to averaging effects, which yields underestimated cloud droplet number concentrations (CDNC).

The relatively coarse spectral resolution of the aerosol bins may induce unwanted discontinuities in the activation spectrum with increasing saturation ratio due to the particle size discretization. To mitigate these effects, the extended SALSA accounts for the distribution of particle number and mass within the critical aerosol size bin using linearly fitted slopes between the bin centres (Korhonen et al., 2005) with both of the available methods for cloud activation.

Evaporation and deactivation of cloud droplets is accounted for through the resolved condensation, upon which activated aerosol particles are released back to the aerosol bin regime as illustrated in Figure 1. For this to take place, a very simple

diagnostic is used, where subsaturation with respect to water vapour is required and the cloud droplet diameter should be smaller than 50 % of the critical diameter dictated by the properties of the CCN (or 2 $\mu$m at maximum). Together with the representation of collision-coalescence processes by Equations (1)and (2), this enables the model to account for aerosol aging inside the clouds. However, please note that chemical processing of aerosol is not presently included in the extended SALSA.

## 2.1.2 Precipitation

Due to our strategy of describing the cloud droplet distribution based on the dry CCN size, the wet droplet diameter in each bin represents a mean over all activated CCN of the corresponding size. Although the wet droplet size can be expected to be somewhat correlated with the dry CCN size, neglecting the variability in the dry/wet size relatioship is an oversimplification when predicting the mass and number of particles converted to drizzle droplets. Therefore, a type of autoconversion parameterization is formulated. Here, a lognormal distribution (selected because of mathematical simplicity) is assumed to describe the variation of the droplet wet size within each cloud droplet bin. The mode diameter is given by the known bin mean wet cloud droplet diameter and the geometric standard deviation is set as $\sigma_g^{ac} = 1.2$, which results in a relatively narrow distribution and is similar to the values used for the cloud droplet size distribution in the UCLALES with bulk microphysics. Setting a commonly used threshold diameter for drizzle droplets, $d_0 = 50$ $\mu$m, the number and mass concentrations of newly formed drizzle from the cloud droplet bins are obtained as an integral over the lognormal distribution from $d_0$ upwards.

The evolution of precipitation is described with an additional set of size bins (Fig. 1). However, since the growth of the drizzle droplets through collection processes is critical to reach rain drop size and produce realistic surface precipitation rates, the precipitation bins are defined according to the wet drop diameter, different from the cloud and aerosol size bins. While the predicted bin properties are again similar to aerosol and cloud droplets, now the aerosol mass (instead of the mass of water) represents a mean for each precipitation size class. This is in contrast with our emphasis of tracking the aerosol size distribution properties, but is an acceptable compromise, since the number concentration of rain drops is always much smaller than the concentration of cloud droplets or aerosols. Thus, their influence on the shape and chemical composition of the ambient aerosol size distribution upon drop evaporation is not considerably obscured by the averaging effects acting on the properties of the aerosol particles embedded inside the rain drops. The precipitation bins cover the size range from 50 $\mu$m to 2 mm. This range is divided into 7 (currently fixed) sections with strongly non-uniform spectral resolution: up to the diameter of 100 μm (first 3 bins) the bin resolution gradually decreased from 5 μm to 35 μm and above the 100 μm range the resolution decreases from 100 μm to 1 mm.

Collection and scavenging of cloud droplets and aerosol particles by precipitation are treated by the coagulation (Equations 1 and 2) as well. Aerosol particles collected by precipitation accumulates the aerosol mass inside the size bins. Upon evaporation of a drizzle or rain drop, it is assumed that a single particle is released (Mitra et al., 1992) and it is placed in an aerosol bin with mean diameter closest to the released dry particle size. The size of the released particle is obtained simply based on the mass and the bulk density of the aerosol. This adds the contribution of drizzle formation on the aerosol processing in the model, albeit, again omitting the chemical processing.

## 2.2 Coupled UCLALES-SALSA

UCLALES (Stevens et al., 2005) is a large-eddy model based on the Smagorinsky-Lilly subgrid model. In the doubly periodic domain, advection of momentum variables is based on a fourth order difference equation with time-stepping by the leap-frog method. For scalars, simple forward timestepping is used. Prognostic variables in the UCLALES are the three wind components $u$, $v$, and $w$ (with the standard meteorological notation), liquid water potential temperature $\theta_l$ and total water mixing ratio $q_t$, plus some additional prognostic scalars depending on the selected thermodynamic level (e.g. rain water). UCLALES contains three thermodynamic levels, which comprise dry, moist and precipitating thermodynamical models, the latter two of which are based on the saturation adjustment method. UCLALES does not include a description for aerosol. Rather, the microphysical processes are driven by a prescribed cloud condensation nuclei (CCN) concentration, taken to represent the cloud droplet number. The drizzle formation is given by Seifert and Beheng (2001) as

$$\frac{\partial q_r}{\partial t} = k_c q_c^2 x_c^2, \tag{9}$$

where $q_r$ is the precipitation mixing ratio, $q_c$ is the cloud condensate mixing ratio, $x_c = q_c/N_c$, where $N_c$ is the CCN concentration, and $k_c$ is a coefficient taking into account the droplet size distribution width and non-equilibrium effects (Stevens and Seifert, 2008). Sedimentation of the drizzle and rain drops is determined by sedimentation velocity which depends on the diagnosed droplet size according to Eq (6).

Coupling the extended SALSA module into UCLALES yields extensive changes in the thermodynamic core of the model as compared to the version based on bulk microphysics, thus adding a new thermodynamic level (Level 4). With the coupled UCLALES-SALSA, condensation and evaporation of water vapour on cloud droplets, rain drops and aerosols is explicitly computed (Eq 8). Therefore, instead of $q_t$ in case of the saturation adjustment method, Level 4 treats $q_c$, $q_r$ and water vapour mixing ratio ($q_v$) as separate prognostic variables. This allows non-equilibrium conditions with respect to water vapour in UCLALES-SALSA, in contrast to the standard UCLALES. $\theta_l$ is retained as a prognostic variable, which allows simple treatment of the latent heat transfer during moist adiabatic transitions.

UCLALES has an option to calculate cloud interaction with radiation using a four-stream radiative transfer solver (Fu and Liou, 1993). The radiation calculation accounts for the diurnal cycle and takes as an input the total number concentration of cloud droplets and the cloud water content. With UCLALES-SALSA, the total number of droplets and condensate mass are obtained as the sum over the cloud droplet size bins and used to calculate radiative transfer the same way as in UCLALES (the aerosol fields are not coupled with radiation in the current model version).

## 2.3 Technical implementation

UCLALES-SALSA is currently implemented under the Fortran95 standard. Output files are written in NetCDF format. For parallel computing the Message Passing Interface (MPI) library is used and the parallellization strategy is based on two-dimensional horizontal blocking of the model domain.

Since the particle number concentrations as well as the masses of different compounds (aerosol species, liquid water) in each particle size bin constitute a prognostic variable, the number of advected scalars is increased from a maximum of 3 in

UCLALES to $\mathcal{O}(100)$ in UCLALES-SALSA even with a simple sulfate-based setup. This obviously has a strong impact on the computational cost. The model runs at about real-time with a Cray XC30 supercomputer using a decomposition with 8x8 grid points per MPI process. While this is a substantial constraint on the applicability of the model, short 12-24 hour (model time) simulations are still easily performed and in the following sections we will show that the presented methods are necessary to

improve our understanding about boundary layer clouds, fogs and aerosols.

## 3   DYCOMS-II

### 3.1   Case description and model configuration

The new UCLALES-SALSA is first configured and tested based on the case DYCOMS-II flight RF02 (Stevens et al., 2003), which took place off the coast of California in July, 2001. The observations conducted in this case featured a mix of open

and closed cell stratocumulus structures, with strong drizzle associated with the former. For the model setup we follow the settings defined by Ackerman et al. (2009): in all simulations, the initial profiles of liquid water potential temperature $\theta_l$, total water mixing ratio $q_t$ (taken as supersaturated vapour in the model initialization process), and $u$ and $v$ wind components were specified with the following equations.

$$\theta_l = \begin{cases} 288.3 \text{ K} & z < z_i \\ 295 - (z - z_i)^{1/3} \text{ K} & z \geq z_i \end{cases} \tag{10}$$

$$q_t = \begin{cases} 9.45 \text{ g kg}^{-1} & z < z_i \\ 3 - 5\left(1 - \exp\left((z - z_i)/500\right)\right) \text{ g kg}^{-1} & z \geq z_i \end{cases} \tag{11}$$

$$u = 3 + 4.3z/1000 \text{ m s}^{-1} \tag{12}$$

$$v = -9 + 5.6z/1000 \text{ m s}^{-1} \tag{13}$$

In the above, $z_i$ is the initial inversion level set at 795 m. In addition, a large-scale divergence of $3.75 \times 10^{-6}$ s$^{-1}$ is assumed, together with prescribed latent and sensible heat fluxes of $93$ and $16$ W m$^{-2}$, respectively.

The simulations span 10 hours. The first hour is considered as the spinup period, during which drizzle formation and all the collision processes are turned off, while cloud activation and condensation processes are active. This prevents spurious effects on the cloud properties during the initial buildup of turbulent kinetic energy (TKE) and settling of the boundary layer properties. The simulation domain spanned 5 km into each horizontal direction and 1600 meters in the vertical, with the topmost 200 meters used as a sponge layer, damping unrealistically reflected gravity waves at the model top. The horizontal

resolution is set to 50 m while the vertical resolution is 20 m. The model uses an adaptive time step, whose maximum value is set to 1 s. During events of strong mixing in the course of the model run, the timestep was occasionally reduced to about $0.5$ s. A more detailed description of the model experiments is given below and their key aspects are summarized in Table 1. The performance of the UCLALES-SALSA model is evaluated by comparing the results with those obtained from similar runs with

UCLALES using bulk microphysics as well as field measurements. This can also be contrasted to the model ensemble used in the LES intercomparison in which UCLALES was a part of (Ackerman et al., 2009). Thus we can isolate and characterize the effects induced by the use of an elaborate sectional microphysical scheme for aerosols and clouds.

### 3.1.1 Reference case experiments

The reference experiments are based on the basic settings in terms of aerosol and cloud microphysics. For the experiment performed with UCLALES-SALSA, designated as LEV4, this means that we use the two-mode lognormal initial aerosol size distribution given in Ackerman et al. (2009), which is assumed to consist of sulphate aerosol. The total number, geometrical mean diameter and geometrical standard deviation are $125 \text{ cm}^{-3}$, 22 nm and 1.2 for the first mode and $65 \text{ cm}^{-3}$, 120 nm and 1.7 for the second mode. In the model startup, the size distribution is remapped into the SALSA aerosol size bins. For comparison with the experiment LEV4, a parallel experiment, designated LEV3, is performed with the UCLALES configuration using bulk cloud microphysics. However, since UCLALES does not contain a description for aerosols, the CCN number concentrations must be prescribed, similar to most other available LES models. In LEV3, the CCN (i.e. cloud droplet) concentration is set to $55 \text{ cm}^{-3}$, which roughly corresponds to the number of cloud droplets initially produced by LEV4 and is also the number used in other LES simulations based on this particular case (Ackerman et al., 2009; Savre et al., 2014).

### 3.1.2 Sensitivity tests

A set of sensitivity tests are performed to further investigate certain aspects of the model. Experiments designated as LEV4HI and LEV3HI are performed. These are similar to LEV4 and LEV3, but with higher aerosol (or CCN for LEV3HI) concentration (mode number concentrations multiplied by 3), and are utilized to study how the coupling between the model microphysics and dynamics reacts to perturbations in the initial aerosol and cloud properties.

## 3.2 Results

### 3.2.1 General features

Figure 2 shows a domain mean time-height plot of the liquid water content (LWC) in the LEV3 and LEV4 experiments. While in the early stages of the simulation the LWC and the macroscopic cloud structure are quite similar between LEV3 and LEV4, after about 4 hours the results start to diverge substantially, marking a clear shift in the boundary layer dynamics. Whereas the LEV3 simulations maintain a solid stratocumulus deck until the end of the simulated period, LEV4 results in a very thin stratiform cloud deck just below the inversion with low LWC and only 5-10 $\text{cm}^{-3}$ cloud droplets. However, in the last couple of hours of the simulation, this setting is interspersed by occasional cumulus elements with base height around 400 m. The thin filaments below the stratiform cloud shown in Figure 2b are the result of these elements, although they appear weak due to the horizontal averaging (the cumulus clouds were also confirmed from 3-dimensional fields, although not shown here). This is reminiscent of the formation of open cell circulation structures in marine stratocumulus clouds (Wood and Hartmann, 2006), which were also observed during RF02 (Stevens et al., 2003).

Figure 3 shows the total liquid water path (LWP, taken as cloud droplets plus precipitation) and the rain water path for LEV3 and LEV4. Again, the LWP is fairly similar between the two experiments during the first 4 hours and it also agrees quite well with the observed mean LWP, as shown in the figure. After about 4 hours, LEV4 starts to deviate from LEV3. However, in a later stage, a substantial portion of the total LWP is interpreted as precipitation in LEV4, while in LEV3 the mass of precipitation is much smaller. This is mainly due to a diagnostic discrepancy: in LEV4 most of the excess precipitating droplets reside within the cloud layer and partition into the smallest bin, where fall speeds are low and droplets quickly evaporate after descending below the cloud layer. This stems from the details in parameterizing drizzle formation. Differences arise for example from the fact that when large cloud droplets ($> 50\,\mu m$) are considered as drizzle in UCLALES-SALSA, they are transferred to the smallest precipitation bin, beyond which their growth is explicitly modelled (though subject to low bin resolution). Instead in LEV3, a size distribution (based on gamma function) is assumed for precipitation, which causes at least a part of the precipitating droplets to reach surface-reaching rain drop sizes much faster than in LEV4. Figure 4a shows that despite the difference in the rain water path, the surface precipitation rate is of similar order of magnitude between LEV3 and LEV4. The results are also for a large part within the observed range as shown in Ackerman et al. (2009). This is used as the main criterion for setting up the model parameters such as $\sigma_g^{ac}$. Nevertheless, even after considering the differences in drizzle, it is evident, that the boundary layer and cloud properties in LEV4 shift towards a very different state as compared to LEV3.

### 3.2.2 Boundary layer structure

In the LEV4 experiment, the boundary layer shows somewhat more stratified characteristics than that in LEV3. Figure 5 plots the domain mean vertical profiles of potential temperature, water vapour and liquid water mixing ratios. Especially towards the end of the simulation, LEV4 shows a rather distinct division of the boundary layer into two separate mixing regimes. This decoupling of the cloud-driven layer (see for reference the conceptual models e.g. in Harvey et al. (2013)) is evident in both the potential temperature as well as water vapor mixing ratio, with the sharpest gradient taking place around $400\,m$ height, following the criteria defined in Jones et al. (2011). In LEV3 the temperature profile is weakly stable as well after 9 hours of simulation, but less so than in LEV4. In particular, the water vapor mixing ratio in LEV3 does not show the same separation as LEV4.

It is typical for a stratocumulus topped boundary layer to shift towards a decoupled structure in the morning as shortwave heating by the rising sun begins to offset the longwave cloud top radiative cooling and therefore reduces cloud driven mixing (Ghate et al., 2014). Although wind shear may also affect the entrainment and cloud top static stability (Wang et al., 2012), it is not surprising that the sharpest transition in LEV4 occurs around 5 hours into the simulation, which is also close to sunrise at the assumed location. It is also noted that after the initial shift, the decoupled structure is subject to positive feedbacks as it reduces the supply of moisture from the surface to the cloud layer, which further reduces the cloud top radiative cooling and thus the cloud driven mixing. This also weakens the cloud top inversion, allowing transport of heat to the upper mixed layer by entrainment. Due to the stable layer at the decoupling interface, heat transferred to the cloud-driven layer is not efficiently mixed, resulting in even more pronounced decoupling of the cloud layer. By the same argument, a larger portion of moisture

released from the surface by the latent heat flux is confined to the surface layer, thus contributing to the relatively high water vapour mixing ratio in LEV4 as compared to LEV3.

### 3.2.3 Role of microphysics and drizzle

Even though LEV3 and LEV4 simulations are subject to identical external forcings, LEV3 does not show as abrupt changes in the simulated cloud layer nor the boundary layer structure as LEV4 does, indicating that something makes the LEV4 boundary layer more susceptible to undergo the decoupling process. As discussed next, the reason for the initial perturbation towards this different state can be traced back to the representation of microphysics and precipitation.

Figure 4 shows the surface precipitation rate in LEV3 and LEV4 simulations as well as the rate of removal of sulphate aerosol embedded inside precipitating droplets in UCLALES-SALSA, illustrating the model's ability to resolve the aerosol wet scavenging process. The UCLALES-SALSA performs this task with very high detail: the size distribution of aerosols is preserved through activation scavenging after which droplet growth and subsequent drizzle generation favors large soluble particles. The aerosol scavenging by cloud activation is clearly visible as a reduction in aerosol number concentration in LEV4 already during the model spinup as shown in Figure 6. After a couple of hours of precipitation formation in LEV4, the consequences of aerosol scavenging by drizzle and rain fallout become visible in the below-cloud layer as well. Scavenging by precipitation is treated as a coagulation process between the rain drops and aerosols both in the in-cloud and below-cloud layers. Upon collision, the mass of the aerosol particle is moved to the rain drop bin in question, and removed from the aerosol bin along with the corresponding number concentration. The reduction in the number of potential CCN indirectly supports continuing production of drizzle through reduced competition for water vapour between cloud droplets. Eventually, drizzle covers a considerable fraction of the total droplet concentration within the stratiform cloud layer. The scavenging of particles and the reduction in cloud water content due to drizzle start to weaken cloud top radiative cooling already during the first few hours of the simulation in LEV4. In contrast, in LEV3 such a transition towards a thinner cloud layer with lower cloud water content does not take place, because of the lack of representation for aerosol scavenging. This marks a distinct change in the interpretation of the model, since the use of prescribed CCN concentration in LEV3 implies an infinite supply of particles advected to the domain. For LEV4 this is not true and in the absense of emissions (aerosol emissions are not implemented in this model version) the aerosol is gradually depleted by scavenging, which is more reminiscent of the model domain moving with the flow (Lagrangian modelling approach).

Because of the low aerosol concentration and sufficient amount of water available in the model initial state, the presented case favors considerable drizzle production. Considering the detailed description of particles and the aerosol removal mechanisms included in the model, the results shown here are not scientifically surprising, but are used to demonstrate the model's ability to reproduce the transitions in boundary layer and cloud structure due to microphysical interactions. Interestingly, the reduced cloud water and the drizzle maintained by the depletion of aerosol, as shown by Figure 3 for LEV4, resemble the corresponding effects shown by (Yamaguchi and Feingold, 2015) for low aerosol simulations performed with a cloud resolving model. However, UCLALES-SALSA provides the means for more detailed investigations of the impact of the particle size distribution and composition on cloud dynamics and aerosol-cloud interactions, which justifies the added complexity and com-

putational demand. This is demonstrated by Figure 7, showing the size distributions of activated and non-activated particles for two heights right after the 1-hour spinup and after 8 hours into the simulation. It clearly shows the impact of activation on the large diameter end of the distribution in the beginning of the simulation, as well as the fact that the total distribution (activated plus non-activated particles) in the cloud layer corresponds well to the dry aerosol distribution at lower levels. After 8 hours of simulation, the depletion of activation sized particles is evident as well, together with a small increase in coarse mode particles at low levels due to particles released from evaporating drizzle and rain drops. An additional example is given by Figure 8, showing the relative change in particle number concentrations for individual bins averaged over the in-cloud and below-cloud layers. The effects of cloud activation and scavenging by drizzle and rain are clearly seen here as well. In particular, the increase of large particles in the below-cloud layer due to rain evaporation is seen here much more clearly than in Figure 7.

Since UCLALES-SALSA includes a variety of processes that directly influence the size distribution and composition of aerosol particles, this also affects the distribution and variation of the mass of soluble material inside cloud droplets. This, at least in the initial phase of droplet formation, contributes to their growth rate which may affect precipitation formation. In this context, the detailed description of the evolution of the aerosol size distribution provided by the model also enables the investigation of aerosol particle emissions, e.g. giant sea salt particles, and their influence on cloud properties and precipitation.

### 3.2.4   Impact of initial particle concentration

Since it is apparent that drizzle formation and the subsequent impacts of particle scavenging yield the divergence of results between the LEV3 and LEV4 simulations, it is necessary to test how changing the particle number concentrations affects the results. This is done simply by repeating the LEV3 and LEV4 experiments with particle concentrations multiplied by three, designated as LEV3HI and LEV4HI. With higher particle concentrations, the precipitation reaching the surface is very small in both simulations, which suppresses the wet scavenging effect in LEV4HI. As a result, the cloud properties in LEV4HI remain quite close to those in LEV3HI during the simulated period. This is seen in the domain mean profiles of LWC and the boundary layer thermodynamical properties, shown in Figure 9, which are indeed remarkably similar between the two experiments. This shows that in conditions where the additional processes and interactions in the new UCLALES-SALSA are not dominating the boundary layer and cloud evolution, the results remain physically consistent with the more simple model versions. It should be noted though, that if the LEV4HI simulation would be continued over an extended period of time, the supply of moisture by the (constant) latent heat flux and the effects of cloud processing on the aerosol size distribution would eventually create drizzle and rain, which would then lead to a similar situation as seen in the experiment LEV4. It has been shown that maintaining a steady-state cloud structure requires aerosol replenishment from multiple sources, including aerosol emissions (Wood et al., 2012). Although there is some aerosol replenishment through mixing from the free troposphere in our model experiments, this is not enough to maintain the cloud deck over prolonged periods of time. Considering the outcomes of the experiments LEV4 and LEV4HI, the model results here are consistent with the findings of Jiang and Wang (2014) regarding the effects of aerosol replenishment. Thus, implementation of aerosol emissions into UCLALES-SALSA is part of our future plans.

## 4 Simulating fog formation and evolution

### 4.1 Case description and model configuration

To demonstrate the versatility of UCLALES-SALSA, the model is configured according to the conditions from a radiation fog event that took place at the UK Met Office research site at Cardington in the night of $12-13^{\text{th}}$ of February, 2008 (Porson et al., 2011; Price, 2011). Simulations using different aerosol concentrations and horizontal wind profiles are described to illustrate the potential effect of aerosol and wind shear on the properties of radiation fog.

In addition to adapting the model initial conditions (temperature, humidity and wind profiles, aerosol size distributions) for this particular case, the main differences in the model configuration here, as compared to the DYCOMS-II case in Section 3, have to do with spatial and temporal resolution and the surface forcing. Here, the model is run with a very high resolution, vertically spanning $1.5$ m in the lowest $150$ m. Above, the resolution is gradually decreased so that the model top is at approximately $800$ m with the total number of levels being 165. The horizontal resolution is $4$ m in each direction and the domain covers an area spanning $320$ m by $640$ m. The timestep is set to $1$ s as in the stratocumulus case. However, the adaptive timestep reduces down to around $0.2$ s during the simulation. This is somewhat less than the minimum adaptive timestep length in the stratocumulus case, which is expected due the higher horizontal resolution used here.

In contrast to the experiments in Section 3, the surface heat fluxes are not prescribed, but are determined with a simple parameterization for soil energy balance, which is coupled with the radiation scheme (Ács et al., 1991). Moreover, the scheme accounts for the heat transfer between surface and deeper soil. For the latent heat flux the surface is assumed to be saturated with respect to water. This can be assumed to be a fairly good approximation until the fog dissipation phase, when the evaporation from the warming surface can deplete the water from the surface layer and the assumption of a saturated surface may overestimate the latent heat flux. Due to the simplicity of the surface energy balance model, to produce reasonable surface cooling rates with respect to observations (Price, 2011), the equation for surface heat capacity was tuned to yield values on the order of $1000$ J kg$^{-1}$ K$^{-1}$, which also depends on the soil water fraction.

The settings for microphysics in the UCLALES-SALSA run are kept similar to those used in the DYCOMS-II case (Section 3), with the exception that drizzle formation is switched off in the fog simulations. This is justified due to the fact that the liquid water content in the fog remains relatively low and the sedimentation of cloud droplets is the main sink of cloud water. This setting also conforms with the model setup by Porson et al. (2011). In addition, while droplet number concentrations were prescribed in the simulations performed by Porson et al. (2011), in UCLALES-SALSA the droplet activation is computed based on the growth of the aerosol particles to sizes larger than their critical diameter at the water vapour supersaturation, which is resolved by the model. While this method for cloud activation was also used in the experiments in Section 3, it is particularly important here, since in radiation fogs the droplet formation is mainly driven by the radiative cooling at the top of the fog layer. Solving the condensation equation (Eq 8) also allows the evaporation of cloud droplets inside the fog if the water supersaturation falls below the equilibrium saturation ratio in the smallest cloud droplet bins. This process can reduce the number of droplets and has been found to take place also in clouds (Wood et al., 2002; Romakkaniemi et al., 2009). Note,

that no spinup period in terms of the configuration of the microphysical processes is used since here it generally takes a few hours from the start of the model run for the fog to emerge.

### 4.1.1 Experiments

The impact of aerosols on fog formation is first investigated by three parallel experiments with zero initial horizontal wind velocities, which differ in their initial particle concentration. As the information about aerosol concentration is not available for this study, we use a bimodal aerosol size distribution with mean sizes of $50$ nm and $150$ nm. In all simulations the number concentration in the Aitken mode is kept at $1000$ cm$^{-3}$ while the number of accumulation mode particles is increased consecutively so that the accumulation mode particle concentrations are $200$, $400$ and $800$ cm$^{-3}$ in experiments A200, A400 and A800, respectively. An additional experiment, A400W is then presented, where the model is initialized with the horizontal wind data from (Porson et al., 2011). The list of experiments is summarized in Table 2.

### 4.2 Results

Similar to the observation-based reports by Price (2011) and LES studies (Nakanishi, 2000; Porson et al., 2011), the fog layer investigated here undergoes distinct thermodynamical transitions during its evolution. Initially, the fog forms near the surface in a very stable layer due to the longwave cooling effect. As the fog-layer encroaches upwards and more droplets are activated at the fog top layers, its optical thickness increases which reduces the radiative cooling effect at the surface. At the same time the peak of radiative cooling at the fog top region becomes more pronounced. Figure 10 shows the evolution of the fog droplet concentration (sampled at 10 meters height) and the growth of the fog layer thickness. For the experiments A200, A400 and A800 (initialized with zero horizontal wind) the increase in the number of droplets due to the increasing aerosol concentration is clearly seen. A higher initial aerosol concentration yields an increased fog layer depth, but the differences between the experiments are minor. This is due to the stability of the temperature profile, which suppresses the mixing especially with low aerosol concentrations, as show in Figure 11.

In the early morning there is a transition from stable to almost neutral temperature stratification inside the fog (Fig. 11). Higher aerosol concentrations promote increased optical thickness of the fog layer, which leads to faster formation of the neutral temperature profile. This is qualitatively similar to the results presented in Price (2011) and is attributed to the reduction in the surface longwave cooling effect with optically thick fog layers and to the supply of heat from the soil. As can be seen from Figure 10, the earlier formation of a neutral temperature profile with higher aerosol load further enhances the aerosol effect on fog droplet concentration (04 UTC in A800) through a positive feedback similar to what has been found to take place at the top of fog, where the increase in droplet concentration enhances radiative cooling which again feeds back as a higher supersaturation and enhanced particle activation (Maalick et al., 2016).

Figure 12 shows the profiles of radiative cooling rate and the water vapour supersaturation as a function of time for the three experiments. As expected, the peak radiative cooling is indeed found near the top of the fog layer. Moreover, the intensity of the cooling increases with increasing aerosol concentration, owing to the higher optical depth: in A800 the peak cooling rate is approximately $7$ K hr$^{-1}$ and in A200 $4$ K hr$^{-1}$. This is in agreement with the range of values reported in Nakanishi (2000).

The peak water vapour supersaturation is found at the same altitudes as the strongest radiative cooling. However, as larger particle concentrations deplete the available water vapour more efficiently, the highest supersaturations occur in the experiment with the lowest particle concentration (A200).

These findings illustrate the ability of UCLALES-SALSA to provide a realistic description of not only the thermodynamic and microphysical properties of fogs, but also the aerosol-fog-radiation interactions and feedbacks on the dynamics. The results from the experiments A200, A400 and A800 compare quite well with those reported in Porson et al. (2011). This includes the rate of growth of the fog layer depth, despite the fact that their simulations were initialized with non-zero horizontal wind profiles. However, the growth rate is considerably lower than in the observations, where the fog top reaches about 100 m within 7 or 8 hours from the first appearance of the fog (see figure 5 in Porson et al., 2011). For UCLALES-SALSA, this is presumably because of the lack of shear generated turbulence. Wind shear has been shown to be very important in controlling the turbulence characteristics inside radiation fogs (Bergot, 2013). Thus, in the additional experiment A400W, UCLALES-SALSA is initialized with an approximately similar wind profile as in (Porson et al., 2011). Interestingly, in this case the growth of the fog layer corresponds much more closely to the observed, as shown by the dashed line in Figure 10. The wind shear present in A400W (Figure 13) yields vertical mixing, which strongly enhances the droplet production within the fog layer even at the initial phase (Figure 10). The mixing and perturbations in radiative heating, as compared to the zero-wind experiments, produce the neutral temperature stratification quite quickly and the strength of the inversion at the top of the fog is also slightly reduced, as shown by Figure 13. This allows more rapid growth of the fog layer, the depth of which reaches over 150 m by morning. This is even deeper than suggested by the observations, and can be attributed to e.g. missing advection effects or possible differences in the initial moisture or temperature profiles. At the same time the increased mixing enhances droplet activation and decreases the differences caused by changing the initial aerosol concentration (not shown).

The results point towards the importance of a detailed representation of the microphysical processes in cases of fog formation. In particular, the size-resolving microphysics in UCLALES-SALSA result in a peak number concentration in the fog droplet size distribution at approximately 25 µm in terms of the wet diameter, which agrees with the observed range between 20 µm and 25 µm based on the measurements presented in Price (2011). This has many positive implications, since realistically capturing the droplet growth is important for representing the droplet sedimentation, which is an essential driver for the fog evolution. The droplet number concentration in the experiment A400 (Figure 10) agrees quite well with the observed range (20 cm$^{-3}$ to 60 cm$^{-3}$) illustrated in Porson et al. (2011) as well before the fog layer rises to form low level stratus in the morning. Similar values are also seen for A400W in the development phase before midnight. However, after midnight the droplet number concentration increases substantially due to activation of even smaller particles as the mixing intensifies because of the wind shear and reduced stability. The resulting high droplet concentrations owe at least in part to the fact that detailed information about the aerosol size distribution was not available. Nevertheless, it is clear that these microphysical aspects are directly linked to the fog and boundary layer dynamics. An increased fog optical depth due to an increased droplet concentration will delay fog evaporation in the morning after sunrise, which thus couples the aerosol concentration with fog occurrence. However, to fully evaluate the aerosol effect on fog lifetime, a more detailed land surface scheme is needed to correctly simulate the latent heat flux and atmospheric water content after sunrise.

## 5 Conclusions

The implementation of a novel bin-microphysics scheme for aerosol, clouds and precipitation in an LES model was presented. The coupled model is based on well-established components: the UCLALES large-eddy simulation model and the SALSA aerosol model, extended with cloud droplets and rain. The bin system for aerosol and clouds follows a unique approach, where the size bins are defined according to the dry particle size for both activated and non-activated particles in an attempt to hold detailed information about the aerosol size distribution both in ambient air and within clouds. This also enables an elaborate description of the effects of cloud processing through collision-coalescence on the properties of the aerosol population as well as a size and composition-resolved simulation of the wet scavenging of aerosol.

The model was tested and evaluated using two well-characterized cases which have also been simulated with LES models in previous work: one comprising marine stratocumulus clouds from the DYCOMS-II campaign and another based on measurements of a radiation fog event in Cardington, UK. For the stratocumulus experiments, UCLALES-SALSA initially produced very similar cloud and boundary layer properties as other LES model versions, many of which rely on bulk microphysics and prescribed particle or droplet concentrations. However, after about 5 hours, UCLALES-SALSA shifted towards a very different boundary layer state, as compared to the standard version of UCLALES, resulting in a thin stratiform cloud deck at the top of a decoupled layer instead of a solid stratocumulus cloud layer. This shift was attributed to the wet removal of aerosol particles through precipitation, which eventually led to a decrease in cloud droplet number and water content. This enhanced the susceptibility of the boundary layer to undergo a significant decoupling, which was triggered by the change in radiation budged during sunrise, which then yielded even more dramatic shift in the cloud properties, forming a feedback loop. Such behavior was not reproduced by the standard UCLALES nor by most of the models used in Ackerman et al. (2009), which is due to the use of prescribed microphysical properties and the lack of interactions treated by the model. While the transition in the cloud properties simulated by UCLALES-SALSA resembles closed-to-open cell transitions in marine stratocumulus, it is noted that the rather small model domain ($5 \times 5$ km) is much too small to represent the circulation dynamics and feedbacks closely related to the real-world mesoscale morphological transitions. Nevertheless, the results are encouraging and show that the model may very well provide the necessary new information related to aerosol-cloud-precipitation interactions in future studies to explain the observed stratocumulus characteristics.

In another set of experiments, the skill of the model in simulating fog formation and development was shown. The model was able to capture the evolution of the fog radiative properties and the resulting changes in the thermodynamical profiles. While increasing the initial aerosol concentration had only slight impact on the growth of the fog layer depth, larger particle concentrations did clearly affect the rate of evolution of the temperature profile, which showed a transition from very stable conditionsm to an eventually almost neutral profile. This is qualitatively in agreement with the observed behavior (Price, 2011). While the growth of the fog-layer depth was clearly underestimated, as compared to observations, when the model was initialized with zero wind speeds, setting a realistic wind profile resulted in a growth rate very similar to the observations. With horizontal wind present, the formation of a neutral temperature stratification is even more pronounced than with zero wind conditions, and even more resembles the observed properties. Porson et al. (2011) identified advection and drainage flows as

plausible explanations for the discrepancy between their model and observations. The results presented in this study also bear these deficiencies and are also affected by other shortcomings, such as the surface scheme which is most likely over-simplified. The remaining differences between the radiation fog simulated by UCLALES-SALSA and the observations notwithstanding, the results of this study still make a strong point for a very detailed representation of aerosol and cloud microphysics in
simulating the fog evolution.

The need for high-resolution models that can accurately simulate the effects of aerosol-cloud interactions on both aerosols and clouds and couple these effects to the dynamical features of the atmosphere is clearly highlighted by the current challenges e.g. in climate research. UCLALES-SALSA provides these abilities making it a highly sophisticated, yet computationally relatively efficient alternative to investigate the role of aerosol in marine stratocumulus clouds or fogs, or the process of wet
scavenging. Although the model is currently limited to warm clouds only, implementation of ice processes is on the way and will be published in a separate paper. Work is currently done also to add treatment of semivolatile aerosol species in the model and to couple the aerosol fields with radiation computation. This will extend the repertoire of the model towards more elaborate studies of the aerosol-cloud interactions as well as towards ice and mixed-phase clouds, whose representation in climate models and the deficiencies therein have recently started to attract more widespread interest.

**Code availability**

The model source code and input files needed to reproduce the simulations presented in this paper can be downloaded from Github at https://github.com/UCLALES-SALSA.

*Acknowledgements.* This work was supported by the Academy of Finland (project number 283031 and the Centre of Excellence in Atmospheric Science, 272041), the European FP7 project BACCHUS (Grant agreement no: 603445) and the European Research Council project
ECLAIR (grant 646857). We gratefully acknowledge Prof. Bjorn Stevens for providing the UCLALES code.

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

**Table 1.** The DYCOMS-II model experiments with their key configuration details.

| Experiment | SALSA | Particle number $(\mathrm{cm}^{-3})$* |
|---|---|---|
| LEV3 | off | 55 |
| LEV4 | on | 190 |
| LEV3HI | off | 165 |
| LEV4HI | on | 570 |

*This is the prescribed CCN concentrations when SALSA is not used, otherwise the total initial aerosol number concentration.

Stevens, B., Giorgetta, M., Esch, M., Mauritsen, T., Crueger, T., Rast, S., Salzmann, M., Schmidt, H., Bader, J., Block, K., Brokopf, R., Fast, I., Kinne, S., Kornblueh, L., Lohmann, U., Pincus, R., Reichler, T., Roeckner, E.: Atmospheric component of the MPI-M Earth System Model: ECHAM6. J. Adv. Model. Earth Syst., 5, 146-172, doi:10.1002/jame.20015, 2013

Stokes, R. H., Robinson, R. A.: Interactions in aqueous nonelectrolyte solutions. I: Solute-solven equilibria. J. Phys. Chem., 70, 2126-2130, 1996.

Stolaki, S., Haeffelin, M., Lac, C., Dupont, J. C., Elias, T., Masson, V.: Influence of aerosols on the life cycle of radiation fog event. A numerical and observational study. Atmospheric Research, 151, 145-161, 2015.

Terai, C. R., Bretherton, C. S., Wood, R., Painter, G.: Aircraft observations of aerosol, cloud, precipitation and boundary layer properties in pockets of open cells over the southeast Pacific. Atmos. Chem Phys., 14, 8071-8088, doi:10.5194/acp-14-8071-2014, 2014.

Vautard, R., Yiou, P., Van Oldenborgh, G. J.: Decline of fog, mist and haze in Europe over the past 30 years. Nat. Geosci., 2, 115-119, 2009.

Vié, B., Pinty, J.-P., Berthet, S., Leriche, M.: LIMA (v1.0): A quasi two-moment microphysical scheme driven by a multimodal population of cloud condensation and ice freezing nuclei. Geosci. Model Dev., 9, 567-586, doi:10.5194/gmd-9-567-2016, 2016.

Wang, H., Feingold, G., Wood, R., Kazil, J.: Modelling microphysical and meterological controls on precipitation and cloud cellular structures in Southeast Pacific stratocumulus. Atmos. Chem. Phys., 10, 6347-6362, doi:10.5194/acp-10-6347-2010, 2010.

Wang, S., Zheng, X., Jiang, Q.: Strongly sheared stratocumulus convection: an observationally based large-eddy simulation study. Atmos. Chem. Phys., 12, 5223-5235, doi:10.5194/acp-12-5223-2012, 2012.

Wood, R., Irons, S., Jonas, P. R.: How important is the spectral ripening effect in stratiform boundary layer clouds? Studies using simple trajectory analysis. J. Atmos. Sci., 59, 2681-2693, 2002.

Wood, R., Bretherton, C. S.: Boundary layer depth, entrainment, and decoupling in the cloud-capped subtropical and tropical marine boundary layer. J. Clim., 17, 3576-3588, 2004.

Wood, R. and Hartmann, D. L.: Spatial variability of liquid water path in marine low cloud: The importance of mesoscale cellular convection, J. Clim., 19, 1748-1764, 2006.

Wood, R., Leon, D., Lebsock, M., Snider, J., Clarke, A. D.: Precipitation driving of droplet concentration variability in marine low clouds. J. Geophys. Res., 117, D19210, doi:10.1029/2012JD018305, 2012.

Yamaguchi, T., Feingold, G.: On the relationship between open cellular convective cloud patterns and the spatial distribution of precipitation. Atmos. Chem. Phys., 15, 1237-1251, doi:10.5194/acp-15-1237-2015, 2015.

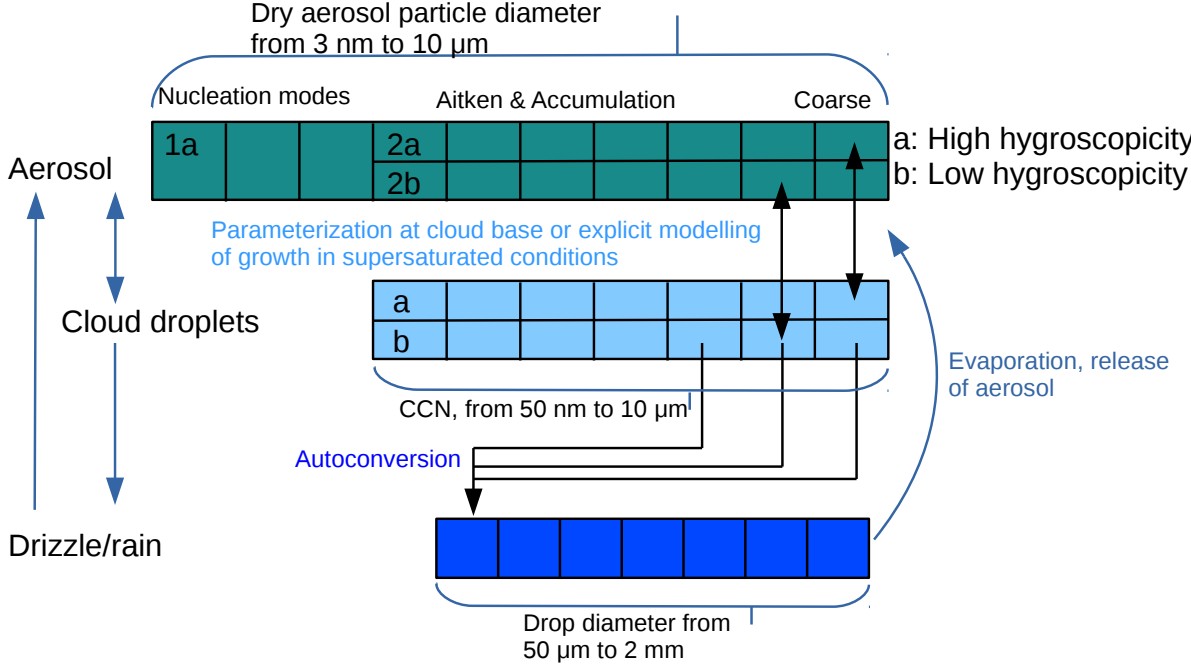

**Figure 1.** Schematic representation of the bin system and processes included in the extended SALSA module. Aerosol bins (green) cover the size range from 3 nm to 10 μm, separated into bin regimes 1a, 2a and 2b (see text). Cloud droplet bins (light blue) are parallel to the aerosol bins in terms of the dry CCN diameter above 50 nm (i.e. the aerosol bin regime 2a/b). Precipitation bins (dark blue), defined according to the wet diameter of the droplet, cover the size range between 50 μm and 2 mm.

**Table 2.** Radiation fog model experiments with their key configuration details. $N_{acc}$ is the number concentration of accumulation mode particles.

| Experiment | $N_{acc}$ $[cm^{-3}]$ | Wind profile |
|:---:|:---:|:---:|
| A200 | 200 | zero |
| A400 | 400 | zero |
| A800 | 800 | zero |
| A400W | 400 | Porson et al. (2011) |

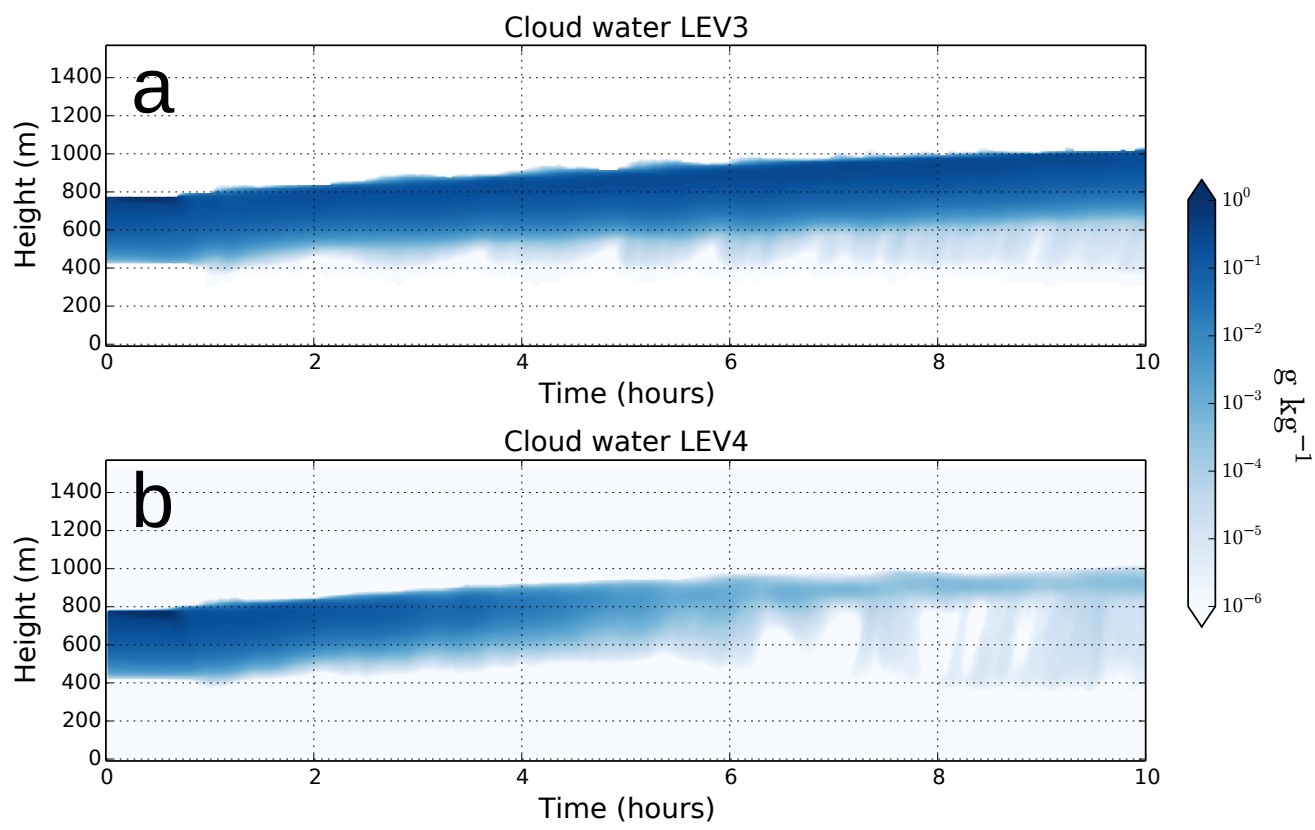

**Figure 2.** Time-height cross section of the cloud water content for LEV3 and LEV4 simulations in $g\,kg^{-1}$.

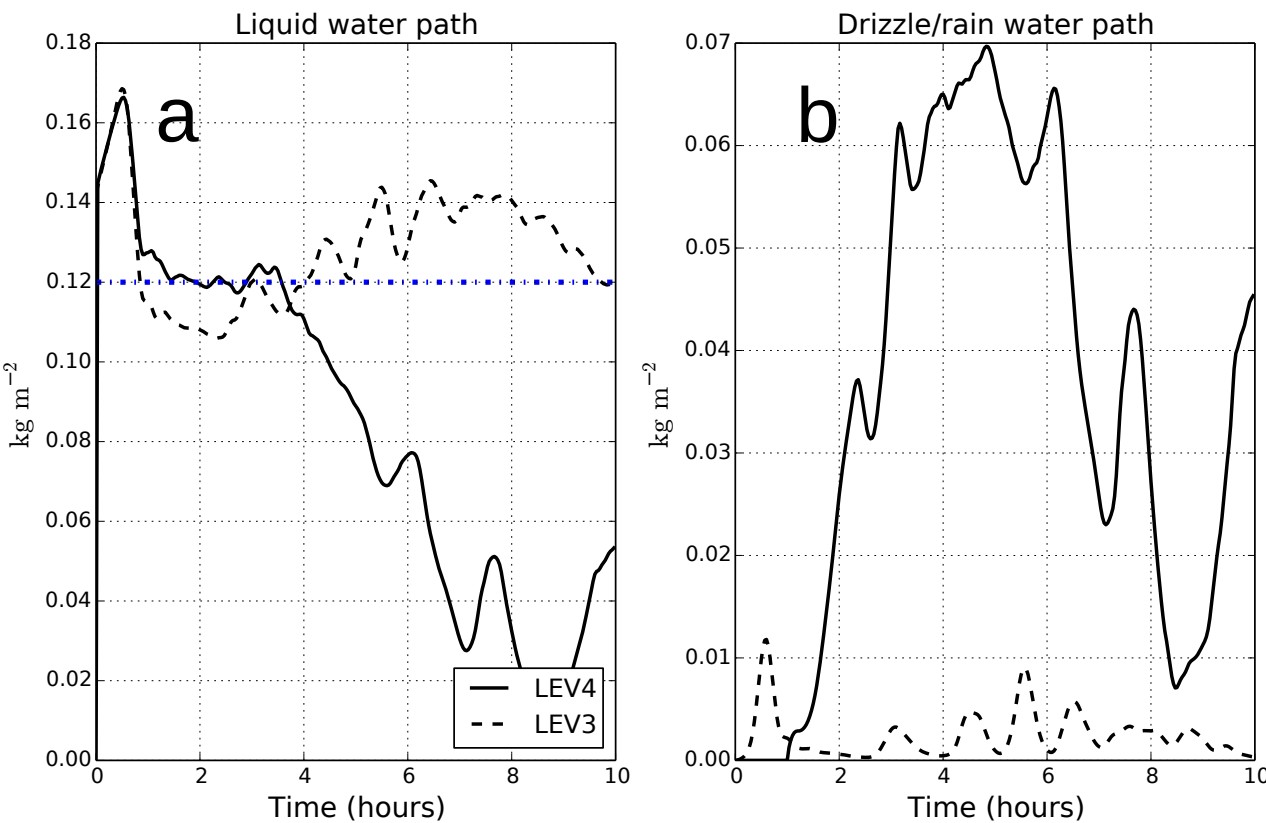

**Figure 3.** a) Liquid water path, interpreted as the total mass of water, including both cloud droplets and drizzle. b) Rain water path, taken as the water mass diagnosed from precipitation bins only. Results from LEV3 are shown with a dashed line while those from LEV4 are shown with a solid line. The horizontal blue dashed line in panel a) represents the observed flight mean liquid water path at $120 \, \mathrm{g \, m^{-2}}$ as reported by Stevens et al. (2003).

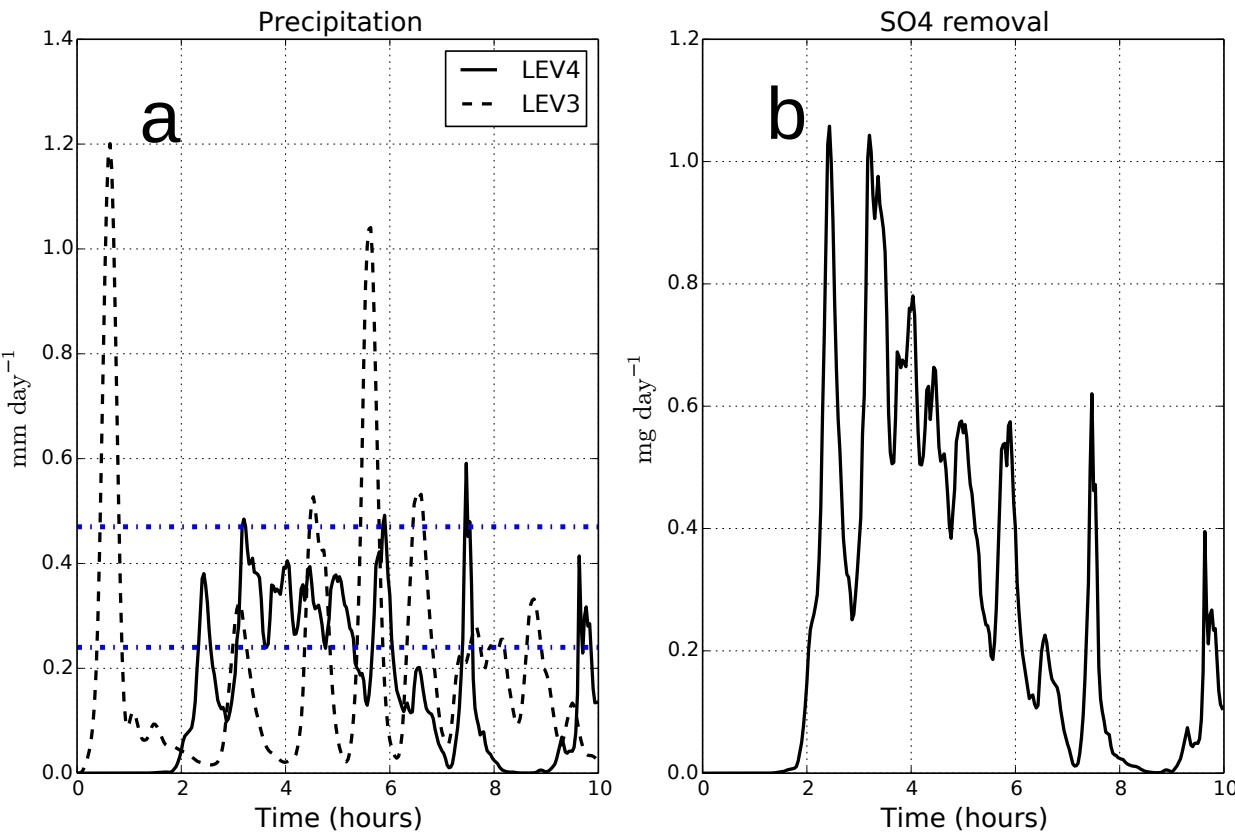

**Figure 4.** a) Surface precipitation rate in $\mathrm{mm\,day}^{-1}$ and b) removal rate of sulphate embedded in precipitating drops in $\mathrm{mg\,day}^{-1}$. Results from LEV3 are shown with a dashed line while those from LEV4 are shown with a solid line. The blue horizontal lines in panel a) indicate range of observed values shown in Ackerman et al. (2009).

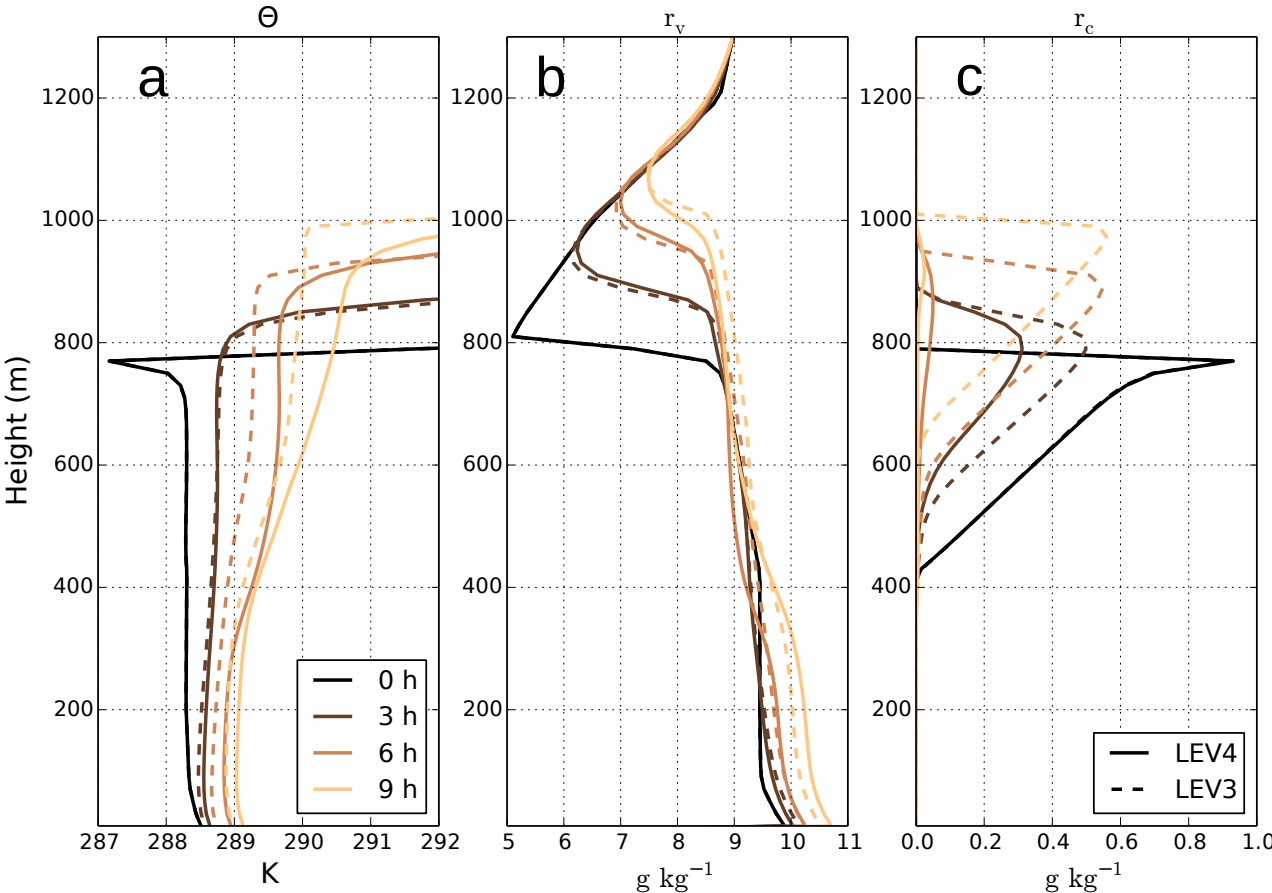

**Figure 5.** Domain mean vertical profiles of a) potential temperature, b) water vapour mixing ratio and c) liquid water mixing ratio. Data is plotted in 3 hour intervals from the initial state of the model to 9 hours into the simulation (from black to orange). Results from LEV3 are shown with a dashed line while those from LEV4 are shown with a solid line.

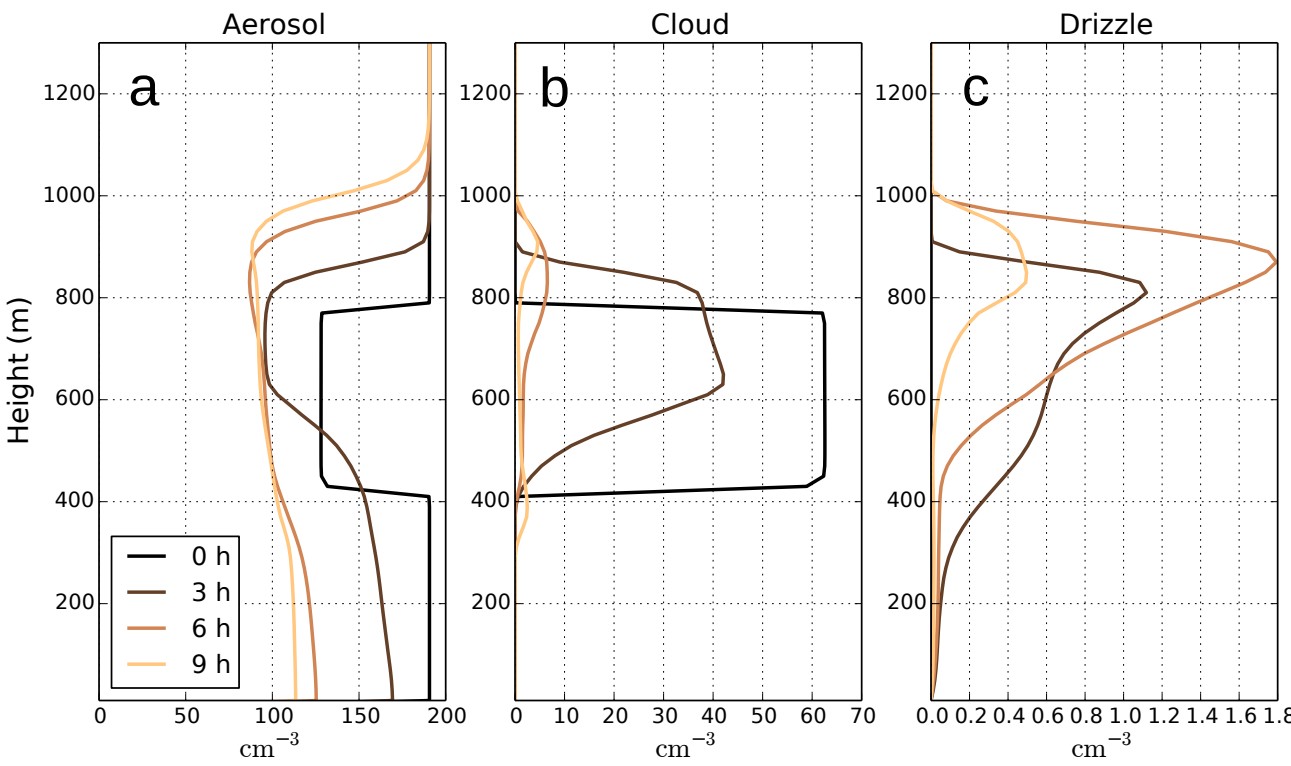

**Figure 6.** LEV4 domain mean profiles for a) aerosol, b) cloud droplet and c) drizzle number concentrations, plotted in 3 hour intervals from the initial state of the model to 9 hours into the simulation (from black to orange).

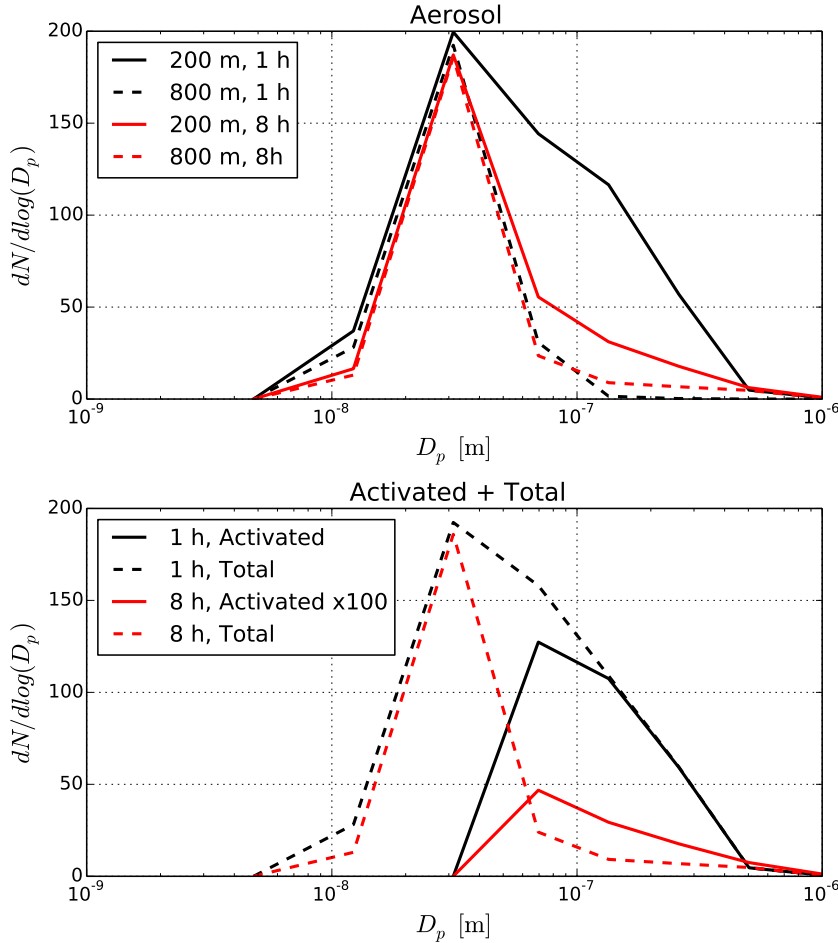

**Figure 7.** Size distributions of a) dry/interstitial aerosol after spinup (1h, black) and after 8 hours (red) averaged over the domain at 200 m (solid) and 800 m (dashed) heights, and b) activated (solid) and total (activated + interstitial) aerosol size distributions after the spinup (black) and after 8 hours (red) sampled at 800 height. Please note that the concentration of activated particles at 8 hours is multiplied by 100 to be visible in the figure. $D_p$ is the particle diameter.

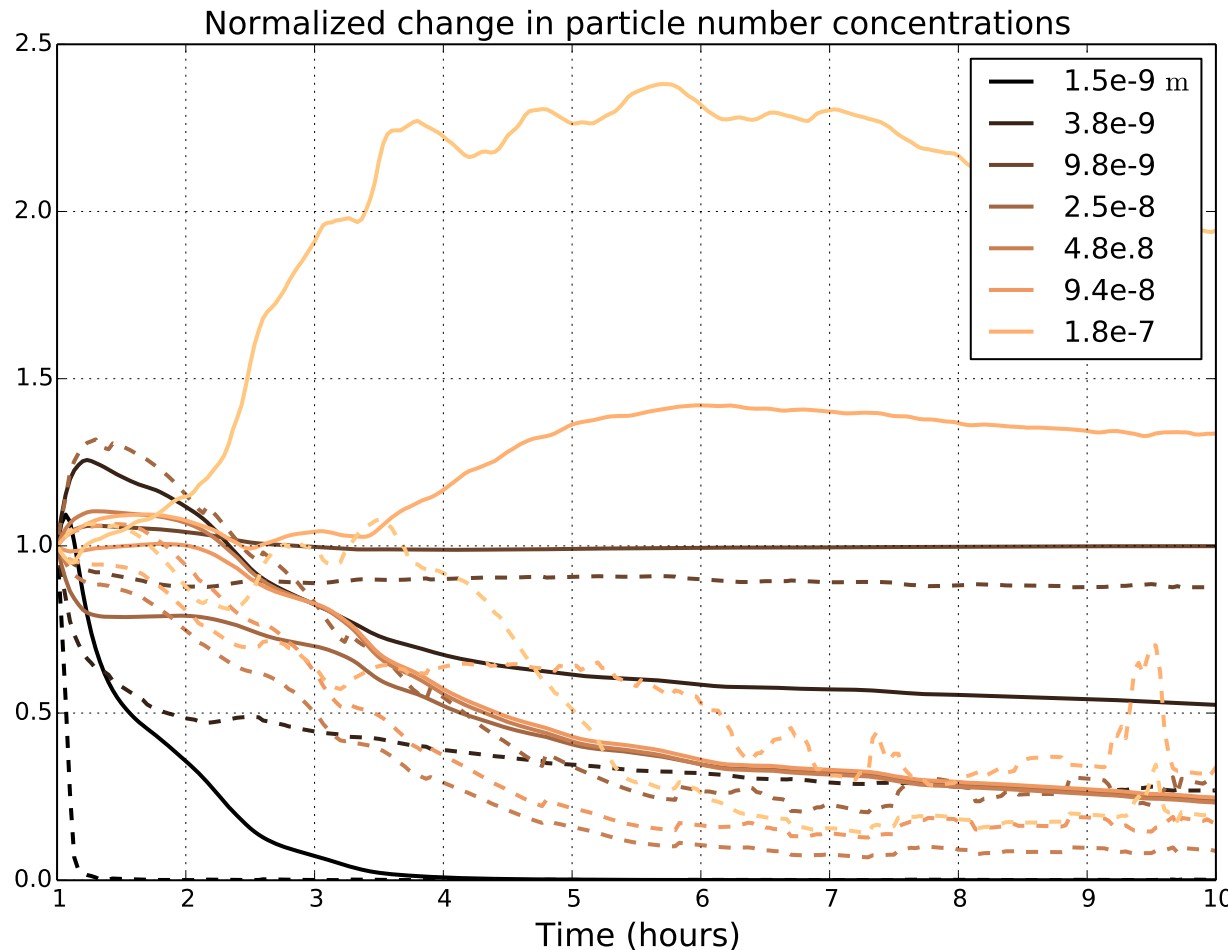

**Figure 8.** Normalized change in particle number concentration in each size bin. The concentrations are presented as a domain average from the i) below-cloud layer (solid lines) and ii) in-cloud (dashed lines). In the latter case, the sum of the number of interstitial particles and activated CCN is presented for each bin. The two largest size bins are not shown because of very small absolute concentrations in this case. The legend gives the lower limit diameter of the presented size bins. Data is show from the end of the spinup period. Number concentrations from this time are used as the normalizing factor for each bin.

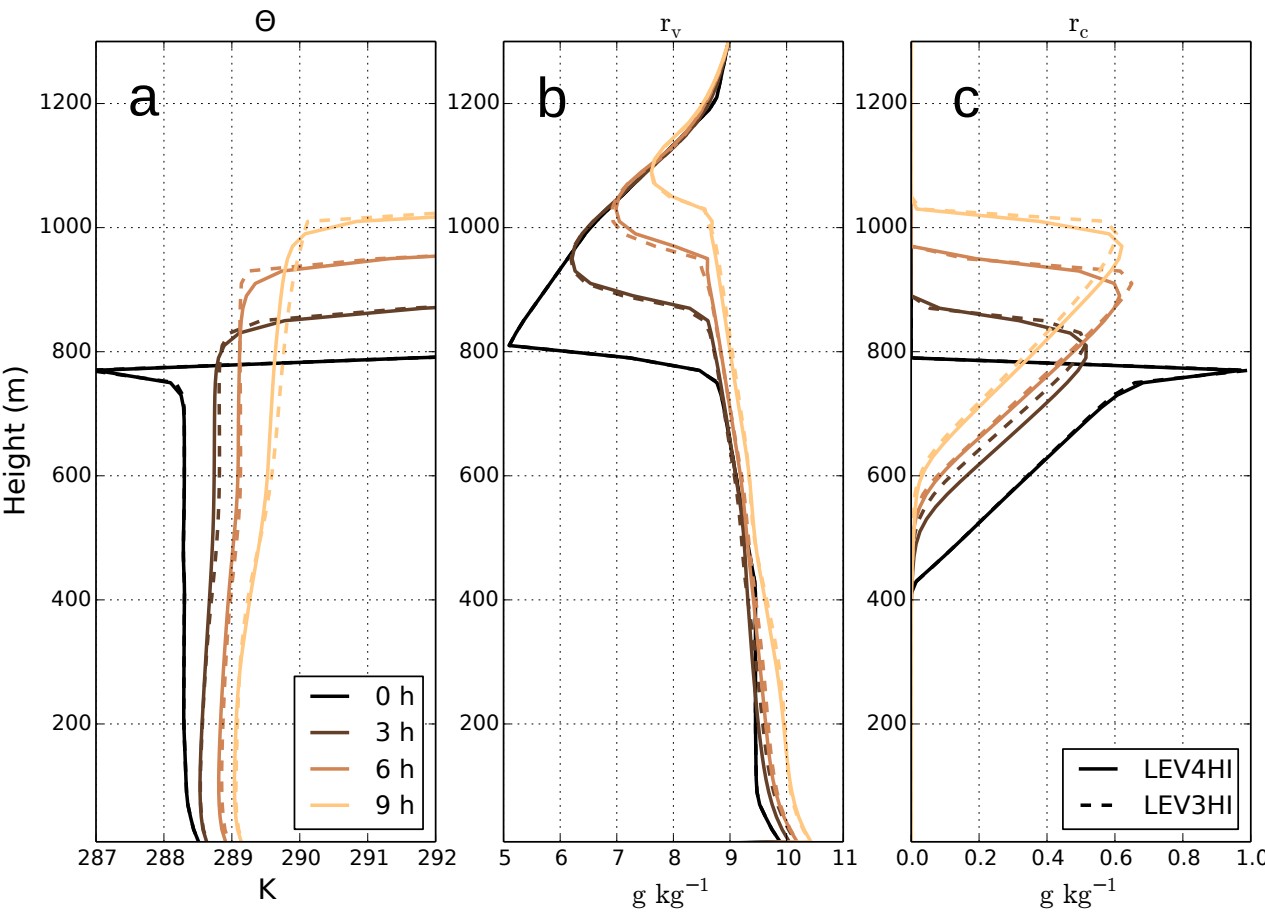

**Figure 9.** Similar to Figure 5, but for the experiments LEV3HI and LEV4HI with high aerosol number concentrations.

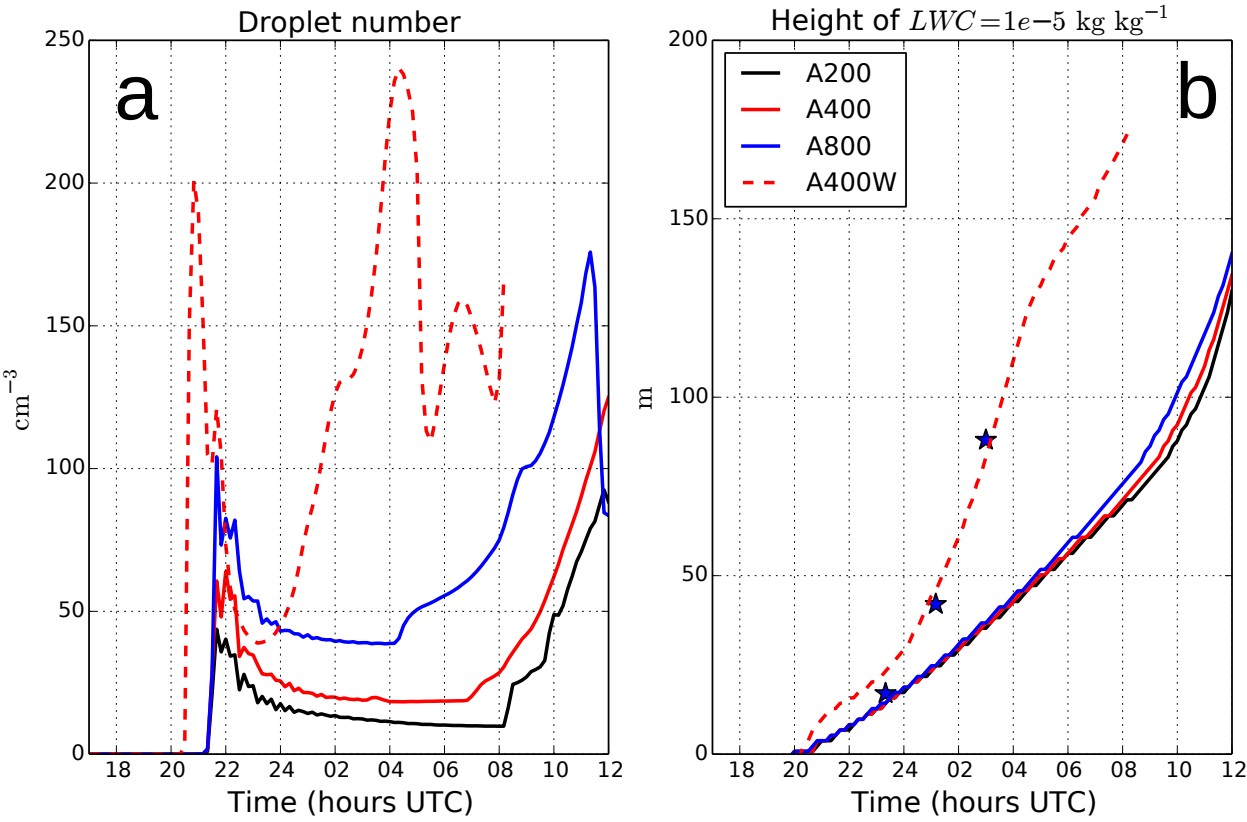

**Figure 10.** a) Fog droplet number concentrations sampled at approximately 10 m height and b) the height of the fog top layer interpreted as the $1 \times 10^{-5}$ kg kg$^{-1}$ isoline for liquid water content. Observed values of the fog layer depth based on tethered balloon data given by Porson et al. (2011) are shown with blue star symbols in panel b).

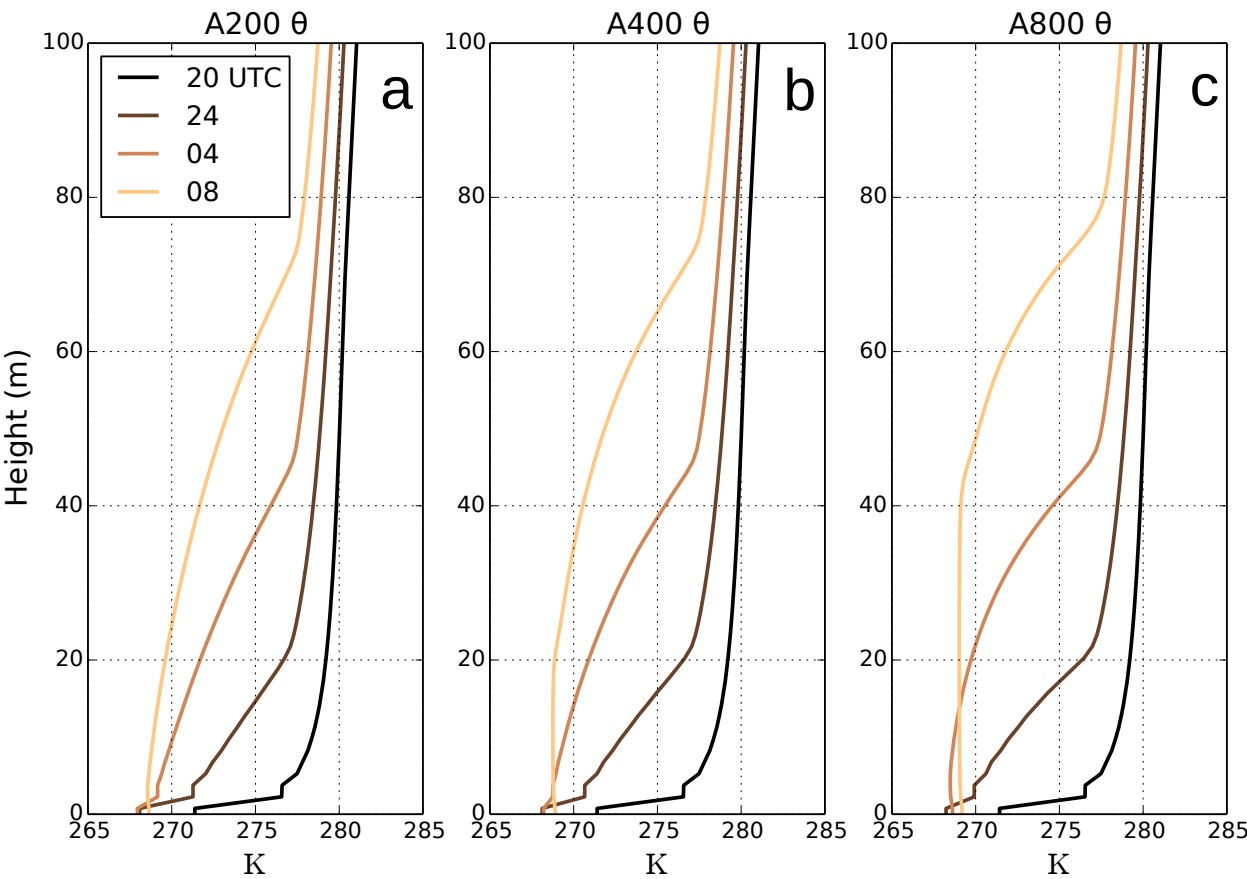

**Figure 11.** Domain mean profiles of potential temperature in 4 hour intervals starting from the formation of the fog layer (from black to orange) for the experiments a) A200, b) A400 and c) A800.

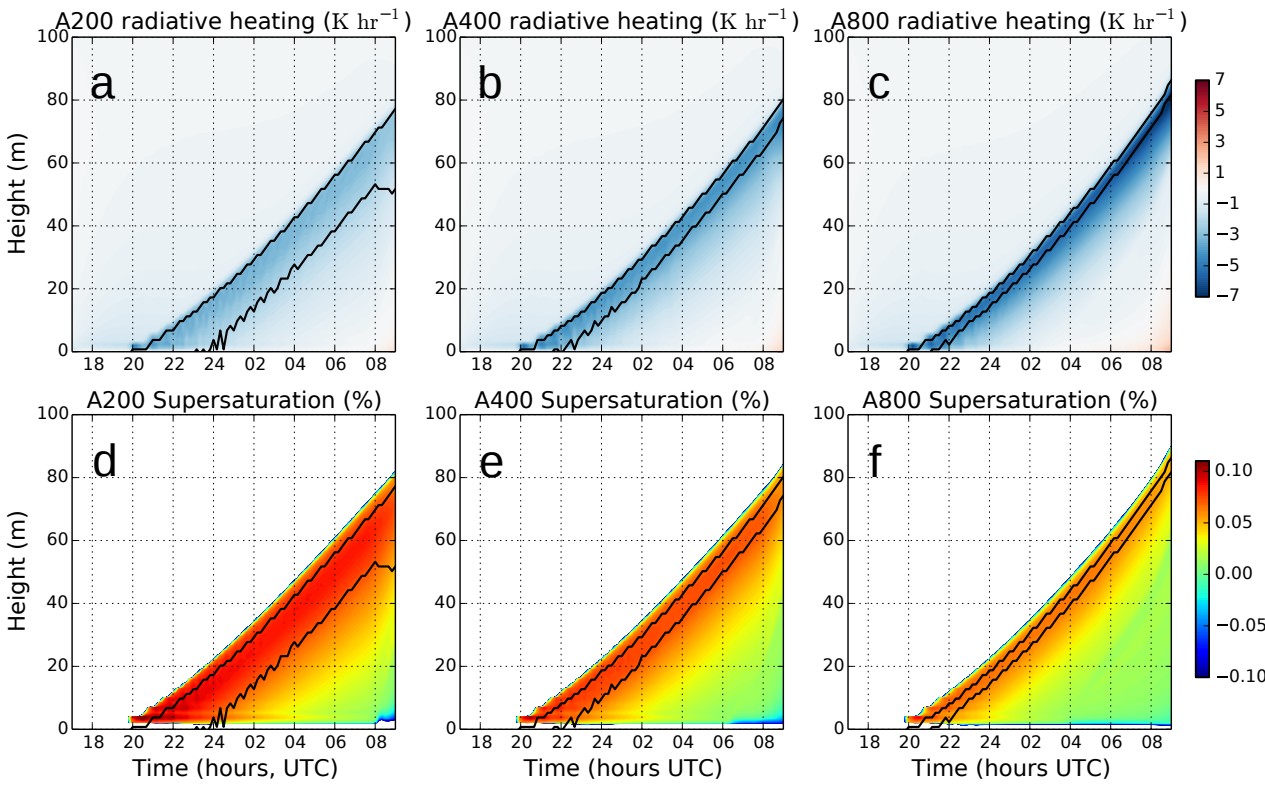

**Figure 12.** a)-c) Radiative heating in $\mathrm{K\,hr^{-1}}$ for the experiments A200,A400 and A800, respectively. d)-f) Water vapour supersaturation in per cent for the same experiments. The upper and lower black curves give the $0.01\ \mathrm{g\,kg^{-1}}$ and $0.1\ \mathrm{g\,kg^{-1}}$ isolines for the liquid water mixing ratio, respectively.

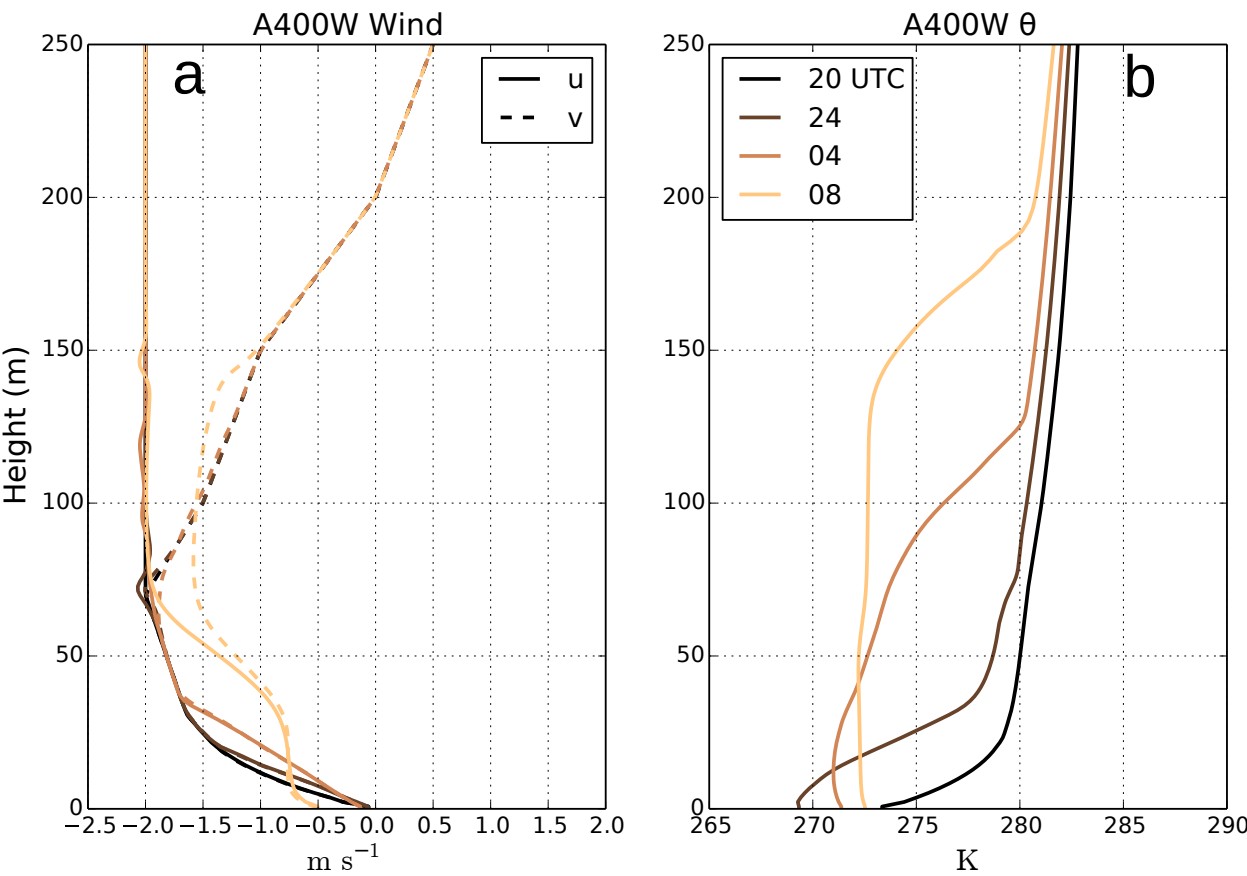

**Figure 13.** a) Domain mean profiles of u and v wind components and b) the potential temperature for the experiment A400W in 4 hour intervals from the formation of the fog layer (from black to orange).