# Peer review of "UCLALES-SALSA v1.0: a large-eddy model with interactive sectional microphysics for aerosol, clouds and precipitation"

_Geoscientific Model Development, 2016_

## Short Comment (SC1) · 7 Jul 2016

Dear authors,

In my role as Executive editor of GMD, I would like to bring to your attention our Editorial version 1.1:

http://www.geosci-model-dev.net/8/3487/2015/gmd-8-3487-2015.html

This highlights some requirements of papers published in GMD, which is also available on the GMD website in the 'Manuscript Types' section:

http://www.geoscientific-model-development.net/submission/manuscript_types.html

In particular, please note that for your paper, the following requirement has not been met in the Discussions paper:

- "The main paper must give the model name and version number (or other unique identifier) in the title."

Please add a version number for UCLALES-SALSA in the title upon your revised submission to GMD.

Yours,

Astrid Kerkweg
* * *

---

## Referee Comment (RC1) · Anonymous Referee #1 · 8 Aug 2016

General Comments:

This manuscript presents a new large-eddy simulation model (UCLALES-SALSA), which includes bins for aerosols, clouds and drizzle. The goal of this model development is to enable better representation of aerosol-cloud interactions, particularly cloud processing and wet scavenging of aerosols. In this study, the UCLALES-SALSA model is used to simulate a marine stratocumulus case and a nocturnal radiation fog case. The results indicate that aerosol-cloud interactions have important influences on the boundary layer dynamics. The manuscript is well written and documents scientifically interesting model developments in the complex field of aerosol-cloud interactions. The manuscript should be suitable for publication provided that the following points can be

satisfactorily addressed. My main concern is that although the simulation results are clearly presented, the manuscript lacks a developed comparison of these results with observations. Confidence in the model developments would be improved with more explicit comparison with observations. As well, there are aspects of the model parameterizations that need clearer description as outlined in the specific comments below.

Specific Comments:

1) P2, L10: To help put the present model developments in context, are you able to point out any previous LES models that have developed similar aerosol-cloud couplings? The text only mentions that such aerosol-cloud schemes are sparse.

2) P3, L19: The sub-range indices 1a and 2a do not appear on Fig. 1 (only a and b are shown, not 1 and 2). Please check this on the figure and check the later references in this paragraph to 2a and 2b. The labels 1 and 2 are confusing because they do not appear on Fig. 1.

3) P4, L3-4: One goal of this work is stated as 'to reproduce the evolution of the aerosol size distribution through cloud processing and wet scavenging by precipitation accurately'. Please consider whether the manuscript would be improved by showing aerosol size distributions. Figure 7 does show the time series of the number concentrations in each bin – would a size distribution figure for hours 0 and 8 be helpful to illustrate the changes? Also please consider showing observed size distributions to improve confidence in the simulations.

4) P4, L8: 'defined to be parallel' – the meaning of this is not quite clear – please clarify.

5) P4, L9-11: ' This way, the properties of the aerosol size distribution are preserved upon cloud droplet activation, as well as evaporation of cloud droplet, though subject to the typical uncertainties inherent in the sectional approach' – please consider rewording this sentence to clarify what is meant by 'properties'.

6) Section 2: Could equations be added to describe the key microphysical processes?

7) P5, L29: Please provide further details about the source for the coagulation kernels.

8) P5, L30-32: How is the dry size of the particle determined when the drizzle drop evaporates? Please clarify.

9) P8, L17: How do you define 'deeper and more massive shallow convection elements'?

10) P8, L32-34: In comparing the LEV3 and LEV4 simulations, it would be helpful to have a clearer description of the parameterization of drizzle formation/loss in LEV3 (the default UCLALES configuration). Perhaps this could be added earlier on in the model description.

11) Fig 4: Where is LEV3 on panel 4b?

12) P9, L33-35: How is scavenging treated in the below-cloud layers? Please consider adding this information.

13) P10 L6: 'lack of representation for aerosol scavenging' – How is aerosol scavenging represented in LEV3? Consider adding this information earlier on in the text to help the reader in understanding these comparisons between the LEV3 and LEV4 simulations.

14) P10, L11-12: 'LWP and rain water path show quite similar features as those obtained with a cloud system resolving model with interactive aerosols' – Please state these 'similar features' more explicitly.

15) Section 3.2: This section includes a detailed discussion of the simulation results for the case DYCOMS-II flight RF02, which was a marine stratocumulus case that took place off the coast of California. Would there be observations available that could be explicitly compared to the simulation output presented here?

16) P11, L22: Why was the drizzle formation switched off for this fog case?

17) P12, L19: Consider adding a table to describe the simulations A200, A400, A800 A400W.

18) P13, L4-5: 'These findings illustrate the ability of the UCLALES-SALSA to provide a realistic description of not only the thermodynamic and microphysical properties....' – Please consider if this statement would be better supported by explicitly showing model-observation comparisons in the manuscript.

19) P 13, L8-9 'growth rate is considerably lower than the observed'...'see figure 5 in Porson et al., 2011' – are there observations that could be explicitly shown here to help the reader understand these comparisons?

20) P13, L21: 'These results point towards the importance of detailed representation of the microphysical processes.' This sentence does not appear to be finished – do you mean in cases of fog?

21) P13, L22: 'UCLALES-SALSA does well' – Are you able to quantify what is meant by 'does well'?

22) P13, L26 'UCLALES-SLASA also agrees well with observations' – again please quantify what is meant by 'agrees well' and consider showing model-observation comparisons in the manuscript.

23) P13, L30: 'a more detailed land surface scheme is needed' – did you test any limiting cases?

24) P14, L29-30 'very similar to the observations'....'even more resembles the observed properties' – Please consider showing these comparisons in the manuscript, likewise showing some model-observations comparisons would be helpful for understanding the model performance for the stratocumulus case.

25) P14, L29: If a realistic wind profile improved the model-measurement agreement – why was the case with winds not used as a default? Did you test A200W and A800W?

Technical Corrections

1) P2,L10: Do you mean 'of' instead of 'off'?

2) P5, L20: 'Evolution of the drizzle droplet population' – should this read drizzle/rain since the upper diameter limit is 2mm?

3) Fig. 3a: Should HI be removed from the legend?

4) P12, L27: Do you mean Fig 8 as opposed to Fig. 9?

5) P12, L31: Consider starting a new paragraph with the start of the Fig. 11 discussion.

6) P13, L13: Do you mean Fig. 9 as opposed to Fig. 10? There is no dashed line on Fig. 9.

7) Fig. 1: What is the meaning of the light blue arrows on the dark blue for the drizzle rain bins? What is the size range for the cloud droplets?

8) Fig 2: Could g kg^-1 be placed beside the color bar?

9) Fig 3: Could drizzle be added to the title of panel b? Also, please check legend for error in simulation names.

10) Fig 7: Please check units on the legend – did you mean m?

---

## Referee Comment (RC2) · Anonymous Referee #2 · 15 Aug 2016

Comments to manuscript gmd-2016-159:
**Introducing UCLALES-SALSA: a large-eddy model with interactive sectional microphysics for aerosols, clouds and drizzle**
by Juha Tonttila, Zubair Maalick, Tomi Raatikainen, Harri Kokkola, Thomas Kühn and Sami Romakkaniemi

**Paper content and a general impression**

The submitted manuscript describes a newly-developed software package for research on aerosol-cloud-precipitation interactions. The presented framework is composed of a modified (adapted) version of the free and open-source Large Eddy Simulation (LES) system UCLALES, coupled with a modified (extended) version of the SALSA aerosol process modelling package. The paper consists of brief description of both pre-existing software packages and of how they were adapted, extended and coupled to result in the UCLALES-SALSA system. Moreover, the capabilities of the developed tool, in particular its applicability to capture aerosol-cloud interactions, are exemplified for two different simulation set-ups.

The topic of the paper matches well with the current interests of the cloud-modelling community – the study fits into an active stream of development of modelling techniques to study aerosol-cloud-precipitation interactions in LES-type frameworks. These concurrent endeavours are not referenced comprehensively in the paper, though.

My main major concern is the misalignment of the paper content with the scope of GMD. The model description itself amounts to ca. 3 pages out of 15 (not counting figures or references), while the rest of the paper deals with the case studies. Of course, this is not the page-count that matters and the case studies do provide valuable examples and validation of the model capabilities. Yet, the model formulation and implementation – in my understanding the key elements of the GMD scope – are clearly described in not sufficient detail. There is not a single equation used in describing the new extensions to the UCLALES and SALSA, even though the authors admit that "*coupling the extended SALSA module into UCLALES yields extensive changes in the thermodynamic core of the model*". A key component newly introduced into the model formulation, representation of cloud droplet collisions and coalescence, is commented with just a single sentence without detailing the numerical method or its implementation. The reader is left without any information about software engineering aspects of the project (without studying the references, the reader would not even learn about the programming language in which the code is written; more importantly such aspects as parallelisation techniques, required environment and tools to use and extend the model are not mentioned and these cannot be guessed). There is no information on how the modified UCLALES and the extensions to SALSA are planned to be disseminated within the community, how existing UCLALES and SALSA users can benefit from the described developments. If the authors would rather prefer to keep the paper focused on the case studies, and not the model formulation and implementation, I suggest submission of a revised version of the text to ACP or similarly scoped journal.

The model code is also not publicly available as of now. It not only makes the paper not compliant with the GMD guidelines, but it also prevents me to fulfil the reviewer's duties – the GMD board clearly states that all papers "*must be accompanied by the code, or means of accessing the code, for the purpose of peer-review*", moreover the journal guidelines "*strongly encourage referees to compile the code, and run test cases supplied by the authors*"[1]. The authors do not detail how the paper readers may reproduce the discussed results.

For the reasons listed above, I am requesting a second round of the review to follow. In my opinion, the manuscript requires substantial changes to reach a good level of readability and to match contemporary standards of research reproducibility pioneered by GMD. Nevertheless, let me repeat that the described research and the developed tool are of prime interest to the community. In particular, the described system is capable of simulating aerosol processing by clouds through activation-collision-deactivation cycles as well as resolving aerosol sources – features not widely available in other LES-type systems.

On the following pages, I provide some more detailed remarks that I hope will help to improve the manuscript.
* * *
[1] Geosci. Model. Dev., 6, 2013. Editorial (10.5194/gmd-6-1233-2013)

**General remarks**

**Few references to other LES aerosol-cloud-precipitation interaction studies**

For the purpose of giving a comprehensive background, as well as of highlighting the unique features of UCLALES-SALSA, I strongly suggest supplementing the list of referenced works with some seminal and/or recent papers on aerosol-cloud-precipitation interaction modelling with LES-type tools. In the list below, I suggest some that might be worth checking. The list is certainly not exhaustive, though:

- Lebo and Seinfeld, 2011: 10.5194/acp-11-12297-2011
- Ovchinnikov and Easter, 2010: 10.1029/2009D012816
- Andrejczuk et al. 2010: 10.1029/2010JD014248
- Shima et al. 2009 10.1002/qj.441
- Feingold et al. 1996: JGR 101 (D16)

In particular, citing some of these works could support or otherwise require rewording of some statements:

- p. 2/line 9/10: "extensive simulations with more detailed and interactive ... schemes ... are relatively sparse"
- p. 2/line 33: "innovative approach" (it is worth clearly stating precisely what is novel here)
- p. 4/lines 6-7: "not a computationally feasible approach"

**Lack of model formulation and implementation details**

As outlined above, I strongly encourage the authors to increase the level of detail in which the model formulation and implementation is described. Here are some examples:

- p. 4/line 20: How the substepping is implemented? Are the grid-mean values kept constant for all substeps within a timestep (in particular, the supersaturation)?
- p. 5/line 13,15: statements seem contrasting: "not knowing the wet droplet diameter exactly" but "bin mean cloud droplet wet diameter" is used, perhaps it is worth summarising clearly what are the variables and constants per bin for each spectrum, and which processes change them – a table would likely give best readability
- p. 5/line 29-30: here the treatment of coalescence is described by just one single sentence without any reference. What are the numerics behind, how the kernels are supplied (if look-up tables, please detail interpolation method)?

**Simulation setup description**

The fact that two contrasting cloud regimes are simulated gives a nice opportunity to pick this a criterion for mentioning or not a given simulation parameter. I suggest thus creating a table listing all model parameters that needed to be changed (or where arbitrarily switched on or off) in order to make the simulation depict fog instead of stratocumulus. This could perhaps allow to shorten a bit the setup descriptions in the text, the initial profile given by eq. 1-4 could then be part of the table (why not just cite the relevant equations in the paper in which the DYCOMS profiles where defined). If adaptive timestepping was used, please provide some statistics on the timestep values for the two different setups. If a spinup period is used for model initialisation, please clearly indicate which processes are on or off for how long, and what are other differences between the spinup and the rest of the simulation.

Also, some of the model features advertised in the beginning of the paper seem not used in the simulations (e.g., condensation of precursor gases and new particle formation mentioned on p. 3/line 30) – please state it explicitly. In contrast, features such as inclusion of the diurnal cycle or the soil energy balance are not mentioned in model description part.

Statements such as "large cloud droplet are considered as drizzle" (p. 8/line 27) or "the surface heat capacity is used as a tunable parameter" (p. 12/line 5) call for numbers.

If I understood correctly, presented simulations lack aerosol sources. In contrast, the setups like DYCOMS-II implicitly assume an infinite reservoir of CCN brought in to the domain by advection. If that is correct, this difference is worth mentioning and perhaps discussing.

**Aerosol processing nomenclature**

Depending on the community "aerosol processing" is associated with different processes if put out of context. Please clearly state, at least in the abstract and introduction, whether chemical processing or collisional processing is addressed. Especially, since condensation of sulphates is mentioned on page 3.

**Section scope**

The introduction section mentions such, distant from the scope of the paper, matters as challenges in climate modelling, arctic temperatures changes, decrease in fog occurrence in Central Europe. For fellow cloud modellers, the links between those topics and the paper scope might be "obvious", for other members of the GMD audience these will seem puzzling, tough. Please either elaborate on how and why these topics are related with the development of UCLALES-SALSA or keep the introduction closer to the paper scope.

The DYCOMS-II section uses up to three-digit section numbering (e.g., 3.2.1) while the fog case is just divided between two case description and Results subsections. I suggest some work on restructuring the two sections to be more similar in both section numbering and, more importantly, the level of detail.

**Specific comments and rewording/correction suggestions**

- Paper-wide:
  - drizzle ⤳ precipitation (in particular in the title, the model is not limited to drizzle and since one of the quantities analysed is the surface precipitation rate, the simulated precipitation is by definition not drizzle)
  - aerosols ⤳ aerosol (e.g., in the title, I don't have a strong opinion on it - just a suggestion)
  - computational burden ⤳ computational cost
  - high computational burden ⤳ resource intensive, etc
  - interactive, fully interactive scheme, interactive description of particles – please explain what you mean exactly (by explaining which models are non-interactive), especially as it is mentioned in the title
  - please ensure that acronyms are explained on first occurrence (e.g., SALSA is only deciphered in section 2.1)
- Abstract:
  - line 1: impacts of ⤳ impacts on
  - line 1,3: improved over what?, more sophisticated than what?, what kind of observations? Please be precise, please try to cater to a wider community, please make sure that the abstract summarises the presented research – global climate and gaps in observations seem not relevant enough to pop up in the very first sentences of the abstract
  - line 4: model, coupled ⤳ model (LES), coupled
  - line 5/6: "microphysical model components" is vague – please state if you refer to SALSA or something else as well
  - line 6: "strategies for ... bin layouts" reads awkward, perhaps the keyword discretisation could help to better convey what is meant? I understand bin layout as a parameter of a given simulation, what is perhaps worth mentioning in the abstract is how the modelled particles are classified and which classes of particles are subject to which processes
  - line 8: "computational cost of the model acceptable" – this is not only subjective but also likely to be objectively false soon
  - line 8: two different cases: one comprising a case with marine stratocumulus ...⤳ two different simulation setups: the DYCOMS-II marine stratocumulus setup
  - line 9-10: It is shown that, in both cases, ...
  - line 13: In radiation fogs, the growth

- line 14: strongly affects ⤳ strongly affect

- Introduction:
  - p. 1/line 18: simulators ⤳ simulations
  - p. 1/line 19: Moeng 1984 – please either use a few references to support the use of LES for decades or cite a recent review paper (preferably)
  - p. 2/line 7: please mention also particle-based models (in addition to bulk, modal and sectional)
  - p. 2/line 10: "mostly due to their high computational burden" – isn't it the multi-scale and multidisciplinary nature of clouds that limits us most and not the computer power?
  - p. 2/line 25: "fogs also feature many different aspects" – please reword
  - p. 2/line 30: four references given to support statement that fogs are affected by anthropogenic emissions – please try to keep balance with the paper and journal scope
  - p. 3/line 6,7: "well-characterised" vs. "findings of" could hint that one is superior to the other, please reword
  - p. 3/line "previous model versions" versioning suggests something linear, in this context we are rather faced with multiple diverging branches, please try to avoid the version when giving a precise version number would be tricky

- Section 2:
  - p. 3/line 16: drizzle/rain ⤳ precipitation
  - p. 3/line 17: since the number 10 is just a setting for a particular simulation, perhaps it is worth explaining instead how the bins are laid out (logarithmically?)
  - p. 3/line 18: what does the Bergman et al. citation refer to? (10 bins?, Figure?)
  - p. 4/line 8: "parallel bins" is hard to understand
  - p. 4/line 15: non-chronological order of citations
  - p. 4/line 16: "very fast relative changes" is vague, isn't it anyhow the stiffness of the governing equations due to presence of multiple size scales that is the crux of the matter?
  - p. 4/line 32/33: "unwanted discontinuities in ... calculation", please reword so it is clear what is discontinuous
  - p. 5/line 8: I suggest removing the last sentence of this paragraph
  - p. 5/line 9: drizzle ⤳ precipitation
  - p. 5/line 12: please rephrase, perhaps referring to statistical moments resolved within each bin would make it clearer
  - p. 5/line 21: please rephrase so that it is clear that wet diameter is the relevant quantity for condensation and collision, currently the sentence suggests that drizzle condensational growth is critical to produce realistic precipitation
  - p. 5/line 22: does SALSA share the implementation of the Abdul-Razzak and Ghan (2002) scheme with some other (open-source?) model?
  - p. 5/line 28: please explain (mathematically) how the bin layout is formulated
  - p. 5/lines 31-32: please refer to the "aerosol processing" in the sentence
  - p. 5/lines 31-32: perhaps citing Mitra et al. 1992 could be used to support the assumption?
  - p. 6/line 10: "default version" might mean something different for each user, please be specific
  - p. 6/line 20: "raising the number of prognostic scalars" ⤳ "increasing the number of advected scalars from O(?) to O(100)
  - p. 6/line 22: please hint the level of concurrency used – otherwise just the computer type makes the statement very vague

- Section 3:
    - p. 6/line 31: is there any limit on the magnitude of the supersaturation during the initialisation?
    - p. 7/line 22: would one of "Reference case"/"Reference setup"/"Reference run" be more apt than "Default case"?
    - p. 7/line 27: from previous statements, I understood model initialisation to be equivalent to the spinup period, here it seems to mean pre-time-zero calculation – please use consistent wording
    - p. 7/line 28: please reserve the word "parallel" for calculation concurrency
    - p. 7/line 28: please define somewhere the "default UCLALES configuration" – again, this might mean different settings to different users
    - p. 7/line 29: "default UCLALES" – does it refer to the "default case", "default configuration" or something else
    - p. 8/line 17: the domain-mean plot discussed was likely not the basis for statements on "massive shallow convective cumulus elements" or open cells; please clearly define where you discuss the figure
    - p. 9/line 20: "By the same token" sounds strange to me (but I'm not a native speaker)
    - p. 9/line 29-31: please reword, "performs this task with very high detail" could be omitted, the use of "beyond" is unclear here
    - p. 9/line 32-33: another example where the reader can be puzzled about differences in spinup and model initialisation
    - p. 10/line 12: please reword "model with interactive aerosols"
    - p. 10/line 24: "to the their" ⤳ "to their"
    - p. 10/line 33: I assume this means very small in one simulation and non-existent in the other, please reword
    - p. 11/line 5: isn't coagulation part of the processing
- Section 4:
    - p. 11/line 28: "water surface pressure" ⤳ saturation vapour pressure?
    - p. 12/line 7-8: "is not available" suggests lack of availability, here it was simply not part of the setup
    - p. 13/line 5: "-radiative" ⤳ "-radiation"
    - p. 13/line 5: if "and feedbacks" is needed, please explain how do you differentiate them from interactions
    - p. 13/line 29: "connect the aerosol concentration into fog existence" – I suggest rewording
- Conclusions:
    - p. 13/line 33: "A new large-eddy simulation model" suggests some new fluid dynamics methodology, while the novelty is elsewhere – please reword
    - p. 14/line 5: please precise what type of processing (i.e., non-aqueous-chemistry related)
    - p. 14/line 27: "observed behavior Price" needs a parenthesis
- References:
    - line 23: korolev ⤳ Korolev
    - line 28: kokkola ⤳ Kokkola
    - line 30: korhonen ⤳ Korhonen
- Figure 2: please sort out the background colour issue in panel b

Hope that helps!

---

## Author Comment (AC1) · 13 Oct 2016

We will add the version number to the topic of the revised manuscript.

Regards, Juha Tonttila
* * *

---

## Author Comment (AC3) · 13 Oct 2016

We thank Reviewer #2 for his/her comments and feedback. The largest changes in the manuscript in response to these comments will be the improved presentation of technical details in addition to the other more specific suggestions given by the Reviewer which will also be implemented in the revised manuscript. Below, we will list all of the Reviewer comments, followed by our response highlighted in *italics.*

**Paper content and a general impression**

The submitted manuscript describes a newly-developed software package for research on aerosol-cloud-precipitation interactions. The presented framework is composed of a modified (adapted) version of the free and open-source Large Eddy Simulation (LES) system UCLALES, coupled with a modified (extended) version of the SALSA aerosol process modelling package. The paper consists of brief description of both pre-existing software packages and of how they were adapted, extended and coupled to result in the UCLALES-SALSA system. Moreover, the capabilities of the developed tool, in particular its applicability to capture aerosol-cloud interactions, are exemplified for two different simulation set-ups.

The topic of the paper matches well with the current interests of the cloud-modelling community – the study fits into an active stream of development of modelling techniques to study aerosol-cloud-precipitation interactions in LES-type frameworks. These concurrent endeavours are not referenced comprehensively in the paper, though.

My main major concern is the misalignment of the paper content with the scope of GMD. The model description itself amounts to ca. 3 pages out of 15 (not counting figures or references), while the rest of the paper deals with the case studies. Of course, this is not the page-count that matters and the case studies do provide valuable examples and validation of the model capabilities. Yet, the model formulation and implementation – in my understanding the key elements of the GMD scope – are clearly described in not sufficient detail. There is not a single equation used in describing the new extensions to the UCLALES and SALSA, even though the authors admit that "coupling the extended SALSA module into UCLALES yields extensive changes in the thermodynamic core of the model". A key component newly introduced into the model formulation, representation of cloud droplet collisions and coalescence, is commented with just a single sentence without detailing the numerical method or its implementation. The reader is left without any information about software engineering aspects of the project (without studying the references, the reader would not even learn about the programming language in which the code is written; more importantly such aspects as parallelisation techniques, required environment and tools to use and extend the model are not mentioned and these cannot be guessed). There is no information on how the modified UCLALES and the extensions to SALSA are planned to be disseminated within the community, how existing UCLALES and SALSA users can benefit

from the described developments. If the authors would rather prefer to keep the paper focused on the case studies, and not the model formulation and implementation, I suggest submission of a revised version of the text to ACP or similarly scoped journal. The model code is also not publicly available as of now. It not only makes the paper not compliant with the GMD guidelines, but it also prevents me to fulfil the reviewer's duties – the GMD board clearly states that all papers "must be accompanied by the code, or means of accessing the code, for the purpose of peer-review", moreover the journal guidelines "strongly encourage referees to compile the code, and run test cases supplied by the authors"[1]. The authors do not detail how the paper readers may reproduce the discussed results.

For the reasons listed above, I am requesting a second round of the review to follow. In my opinion, the manuscript requires substantial changes to reach a good level of readability and to match contemporary standards of research reproducibility pioneered by GMD. Nevertheless, let me repeat that the described research and the developed tool are of prime interest to the community. In particular, the described system is capable of simulating aerosol processing by clouds through activation-collision-deactivation cycles as well as resolving aerosol sources – features not widely available in other LES-type systems.

*As a general statement, we will improve on the description of the model technical details in the manuscript. In the current version, it is stated that the model will be available through Github, and upon request before its release. We are currently still in the process of cleaning up the code for a public release and expect this to be finished shortly.*

**General remarks**

**Few references to other LES aerosol-cloud-precipitation interaction studies**
For the purpose of giving a comprehensive background, as well as of highlighting the unique features of UCLALES-SALSA, I strongly suggest supplementing the list of referenced works with some seminal
and/or recent papers on aerosol-cloud-precipitation interaction modelling with LES-type tools. In the list below, I suggest some that might be worth checking. The list is certainly not exhaustive, though:
• Lebo and Seinfeld, 2011: 10.5194/acp-11-12297-2011
• Ovchinnikov and Easter, 2010: 10.1029/2009D012816
• Andrejczuk et al. 2010: 10.1029/2010JD014248
• Shima et al. 2009 10.1002/qj.441
• Feingold et al. 1996: JGR 101 (D16)
In particular, citing some of these works could support or otherwise require rewording of some statements:
• p. 2/line 9/10: "extensive simulations with more detailed and interactive ... schemes ... are relatively sparse"
• p. 2/line 33: "innovative approach" (it is worth clearly stating precisely what is novel here)
• p. 4/lines 6-7: "not a computationally feasible approach"

*We will adjust the manuscript as suggested by the Reviewer.*

Lack of model formulation and implementation details
As outlined above, I strongly encourage the authors to increase the level of detail in which the model formulation and implementation is described. Here are some examples:
• p. 4/line 20: How the substepping is implemented? Are the grid-mean values kept constant for all substeps within a timestep (in particular, the supersaturation)?
• p. 5/line 13,15: statements seem contrasting: "not knowing the wet droplet diameter exactly" but "bin mean cloud droplet wet diameter" is used, perhaps it is worth summarising clearly what are the variables and constants per bin for each spectrum, and which processes change them – a table would likely give best readability
• p. 5/line 29-30: here the treatment of coalescence is described by just one single sentence without any reference. What are the numerics behind, how the kernels are supplied (if look-up tables, please detail interpolation method)?

*We will increase the level of technical details in the model description, including all the points above raised by the Reviewer. As also requested by Reviewer #1, equations describing the key model processes will be added.*

**Simulation setup description**

The fact that two contrasting cloud regimes are simulated gives a nice opportunity to pick this a criterion for mentioning or not a given simulation parameter. I suggest thus creating a table listing all model parameters that needed to be changed (or where arbitrarily switched on or off) in order to make the simulation depict fog instead of stratocumulus. This could perhaps allow to shorten a bit the setup descriptions in the text, the initial profile given by eq. 1-4 could then be part of the table (why not just cite the relevant equations in the paper in which the DYCOMS profiles where defined). If adaptive timestepping was used, please provide some statistics on the timestep values for the two different setups.If a spinup period is used for model initialisation, please clearly indicate which processes are on or off for how long, and what are other differences between the spinup and the rest of the simulation.

*We will include more detailed information about the model setups as well as the spinup configuration. We will also elaborate on how the setup for fog simulations differs from the stratocumulus case. These differences are mainly comprised of model resolution, surface conditions and the input sounding. Adaptive timestep is used – we will add more detailed information about the values to the manuscript.*

Also, some of the model features advertised in the beginning of the paper seem not used in the simulations (e.g., condensation of precursor gases and new particle formation mentioned on p. 3/line 30) – please state it explicitly. In contrast, features such as inclusion of the diurnal cycle or the soil energy balance are not mentioned in model description part.

*The condensation of aerosol precursors was active in the model simulations, but it's effect was negligible in the current simulations setups. New particle formation was not active, although available. We will mention these and the other features suggested by the Reviewer.*

Statements such as "large cloud droplet are considered as drizzle" (p. 8/line 27) or "the surface heat capacity is used as a tunable parameter" (p. 12/line 5) call for numbers.

*We will add this information in the manuscript.*

If I understood correctly, presented simulations lack aerosol sources. In contrast, the setups like DYCOMS-II implicitly assume an infinite reservoir of CCN brought in to the domain by advection. If that is correct, this difference is worth mentioning and perhaps discussing.

*This is true, as well as the fact that the interpretation of the model simulations indeed does change when switching from the default UCLALES with prescribed CCN concentration to UCLALES-SALSA with a more dynamic description. The latter points more towards a "Lagrangian" simulation in the sense that the depletion of aerosols by clouds and precipitation resembles a domain moving with the flow. We will add discussion of this in the manuscript.*

**Aerosol processing nomenclature**

Depending on the community "aerosol processing" is associated with different processes if put out of context. Please clearly state, at least in the abstract and introduction, whether chemical processing or collisional processing is addressed. Especially, since condensation of sulphates is mentioned on page 3.

*Despite the condensation of aerosol precursors included in the model, for now the model does not have chemical processing. This will be stated more clearly.*

**Section scope**
The introduction section mentions such, distant from the scope of the paper, matters as challenges in climate modelling, arctic temperatures changes, decrease in fog occurrence in Central Europe. For fellow cloud modellers, the links between those topics and the paper scope might be "obvious", for other members of the GMD audience these will seem puzzling, tough. Please either elaborate on how and why these topics are related with the development of UCLALES-SALSA or keep the introduction closer to the paper scope.

*Since this modelling work ultimately aims at providing a research tool to improve the above mentioned features in global and regional climate models, we will keep these topics in the introduction, but will provide more in depth description, as suggested by the Reviewer.*

The DYCOMS-II section uses up to three-digit section numbering (e.g., 3.2.1) while the fog case is just divided between two case description and Results subsections. I suggest some work on restructuring the two sections to be more similar in both section numbering and, more importantly, the level of detail.

*We will break Section 4 into smaller divisions.*

*The specific suggestions below will be implemented into the manuscript unless otherwise mentioned.*
**Specific comments and rewording/correction suggestions**
• **Paper-wide:**
– drizzle → precipitation (in particular in the title, the model is not limited to drizzle and since one of the quantities analysed is the surface precipitation rate, the simulated precipitation is by definition not drizzle)
– aerosols → aerosol (e.g., in the title, I don't have a strong opinion on it - just a suggestion)
– computational burden → computational cost
– high computational burden → resource intensive, etc
– interactive, fully interactive scheme, interactive description of particles – please explain what you mean exactly (by explaining which models are non-interactive), especially as it is mentioned in the title
– please ensure that acronyms are explained on first occurrence (e.g., SALSA is only deciphered in section 2.1)
• **Abstract:**
– line 1: impacts of → impacts on
– line 1,3: improved over what?, more sophisticated than what?, what kind of observations? Please be precise, please try to cater to a wider community, please make sure that the abstract summarises the presented research – global climate and gaps in observations seem not relevant enough to pop up in the very first sentences of the abstract
– line 4: model, coupled → model (LES), coupled
– line 5/6: "microphysical model components" is vague – please state if you refer to SALSA or something else as well
– line 6: "strategies for ... bin layouts" reads awkward, perhaps the keyword discretisation could help to better convey what is meant? I understand bin layout as a parameter of a given simulation, what is perhaps worth mentioning in the abstract is how the modelled particles are classified and which classes of particles are subject to which processes

– line 8: "computational cost of the model acceptable" – this is not only subjective but also likely to be objectively false soon

– line 8: two different cases: one comprising a case with marine stratocumulus . . . → two different simulation setups: the DYCOMS-II marine stratocumulus setup

– line 9-10: It is shown that, in both cases, . . .

– line 13: In radiation fogs, the growth

– line 14: strongly affects → strongly affect

• **Introduction:**

– p. 1/line 18: simulators → simulations

– p. 1/line 19: Moeng 1984 – please either use a few references to support the use of LES for decades or cite a recent review paper (preferably)

– p. 2/line 7: please mention also particle-based models (in addition to bulk, modal and sectional)

– p. 2/line 10: "mostly due to their high computational burden" – isn't it the multi-scale and multidisciplinary nature of clouds that limits us most and not the computer power?

– p. 2/line 25: "fogs also feature many different aspects" – please reword

– p. 2/line 30: four references given to support statement that fogs are affected by anthro-pogenic emissions – please try to keep balance with the paper and journal scope

– p. 3/line 6,7: "well-characterised" vs. "findings of" could hint that one is superior to the other, please reword

– p. 3/line "previous model versions" versioning suggests something linear, in this context we are rather faced with multiple diverging branches, please try to avoid the version when giving a precise version number would be tricky

• **Section 2:**

– p. 3/line 16: drizzle/rain → precipitation

– p. 3/line 17: since the number 10 is just a setting for a particular simulation, perhaps it is worth explaining instead how the bins are laid out (logarithmically?)

– p. 3/line 18: what does the Bergman et al. citation refer to? (10 bins?, Figure?)

– p. 4/line 8: "parallel bins" is hard to understand

– p. 4/line 15: non-chronological order of citations

– p. 4/line 16: "very fast relative changes" is vague, isn't it anyhow the stiffness of the governing equations due to presence of multiple size scales that is the crux of the matter?

– p. 4/line 32/33: "unwanted discontinuities in ... calculation", please reword so it is clear what is discontinuous

– p. 5/line 8: I suggest removing the last sentence of this paragraph

– p. 5/line 9: drizzle → precipitation

– p. 5/line 12: please rephrase, perhaps referring to statistical moments resolved within each bin would make it clearer

– p. 5/line 21: please rephrase so that it is clear that wet diameter is the relevant quantity for condensation and collision, currently the sentence suggests that drizzle condensational growth is critical to produce realistic precipitation

– p. 5/line 22: does SALSA share the implementation of the Abdul-Razzak and Ghan (2002) scheme with some other (open-source?) model?

– p. 5/line 28: please explain (mathematically) how the bin layout is formulated

– p. 5/lines 31-32: please refer to the "aerosol processing" in the sentence

– p. 5/lines 31-32: perhaps citing Mitra et al. 1992 could be used to support the assumption?

– p. 6/line 10: "default version" might mean something different for each user, please be specific

– p. 6/line 20: "raising the number of prognostic scalars" → "increasing the number of advected scalars from O(?) to O(100)

– p. 6/line 22: please hint the level of concurrency used – otherwise just the computer type makes the statement very vague

• **Section 3:**

– p. 6/line 31: is there any limit on the magnitude of the supersaturation during the initiali-sation?

– p. 7/line 22: would one of "Reference case"/"Reference setup"/"Reference run" be more apt than "Default case"?

– p. 7/line 27: from previous statements, I understood model initialisation to be equivalentthe spinup period, here it seems to mean pre-time-zero calculation – please use consistent wording

– p. 7/line 28: please reserve the word "parallel" for calculation concurrency

– p. 7/line 28: please define somewhere the "default UCLALES configuration" – again, this might mean different settings to different users

– p. 7/line 29: "default UCLALES" – does it refer to the "default case", "default configura-tion" or something else

– p. 8/line 17: the domain-mean plot discussed was likely not the basis for statements"massive shallow convective cumulus elements" or open cells; please clearly define where you discuss the figure

– p. 9/line 20: "By the same token" sounds strange to me (but I'm not a native speaker)

– p. 9/line 29-31: please reword, "performs this task with very high detail" could be omitted, the use of "beyond" is unclear here

– p. 9/line 32-33: another example where the reader can be puzzled about differencesspinup and model initialisation

– p. 10/line 12: please reword "model with interactive aerosols"

– p. 10/line 24: "to the their" → "to their"

– p. 10/line 33: I assume this means very small in one simulation and non-existent in the other, please reword

– p. 11/line 5: isn't coagulation part of the processing

• **Section 4:**

– p. 11/line 28: "water surface pressure" → saturation vapour pressure?

– p. 12/line 7-8: "is not available" suggests lack of availability, here it was simply not partthe setup

– p. 13/line 5: "-radiative" → "-radiation"

– p. 13/line 5: if "and feedbacks" is needed, please explain how do you differentiate them from interactions

– p. 13/line 29: "connect the aerosol concentration into fog existence" – I suggest rewording

• **Conclusions:**

– p. 13/line 33: "A new large-eddy simulation model" suggests some new fluid dynamics methodology, while the novelty is elsewhere – please reword

– p. 14/line 5: please precise what type of processing (i.e., non-aqueous-chemistry related)

– p. 14/line 27: "observed behavior Price" needs a parenthesis

• **References:**

– line 23: korolev → Korolev

– line 28: kokkola → Kokkola

– line 30: korhonen → Korhonen

• **Figure 2: please sort out the background colour issue in panel b** *There is in fact no background color issue. In UCLALES-SALSA the condensed water in aerosol particles is classified as "cloud*

*water" which, even though small, becomes visible because of the log-scale of the colormap. We will note this in the caption.*

---

## Author Response (AR1)

We thank Reviewer #1 for his/her constructive comments. The main concern raised by the Reviewer was the lack of direct comparison of the results with observations. We will improve upon this to the extent allowed by the fact that comparing the results from the semi-idealized setup of UCLALES-SALSA with aircraft measurements, where conditions vary significantly even during a single flight leg, is not very straightforward. Below, we will list all of the Reviewer comments, followed by our response highlighted in *italics*

The changes to the manuscript regarding each comment are highlighted in *italics*.

**1)** P2, L10: To help put the present model developments in context, are you able to point out any previous LES models that have developed similar aerosol-cloud couplings? The text only mentions that such aerosol-cloud schemes are sparse.

Aerosol-cloud interactions have been available also in previously published models, but the approach we take to describe the evolution of the aerosol size distribution for both activated and non-activated particles in a bin model is rather unique. To account for this comment, we will add further discussion and references on aerosol-cloud modeling frameworks of similar level of sophistication (e.g. the LIMA scheme; Vié et al., 2016). We will also implement the suggestions by Reviewer #2 concerning the same topic.

We have added this and several other new references to page 2, lines 10-15 of the revised manuscript.

**2)** P3, L19: The sub-range indices 1a and 2a do not appear on Fig. 1 (only a and b are shown, not 1 and 2). Please check this on the figure and check the later references in this paragraph to 2a and 2b. The labels 1 and 2 are confusing because they do not appear on Fig. 1.

We will modify the figure according to the Reviewers suggestions.

The subrange indices are now indicated in Figure 1.

**3)** P4, L3-4: One goal of this work is stated as 'to reproduce the evolution of the aerosol size distribution through cloud processing and wet scavenging by precipitation accurately'. Please consider whether the manuscript would be improved by showing aerosol size distributions. Figure 7 does show the time series of the number concentrations in each bin – would a size distribution figure for hours 0 and 8 be helpful to illustrate the changes? Also please consider showing observed size distributions to improve confidence in the simulations.

Figures showing the simulated size distributions as suggested by the Reviewer will be added. They show that the evolution of the dry as well as the total (activated+non-activated) aerosol size distributions remain consistent and robust with respect to the presented model processes. The initial aerosol conditions used in the model runs are based on the data given by Ackerman et al. 2009, which are somewhat idealized due to the high variability of the aerosol size distributions even within a flight leg. A new figure (figure 7 in the revised manuscript) shows the aerosol size distributions as suggested by the Reviewer. This figure is discussed on page 12, lines 1-6.

**4)** P4, L8: 'defined to be parallel' – the meaning of this is not quite clear – please clarify.

The "parallel bins" in the text refer to setting the cloud droplet bin edges (according to the dry CCN size) identical to those in the corresponding non-activated aerosol bins. i.e. the bins are defined for same dry particle sizes. We have reiterated this in the manuscript.

The "parallel bins" part is explained more clearly on page 6, lines 10-13 in the revised manuscript.

**5)** P4, L9-11: 'This way, the properties of the aerosol size distribution are preserved upon cloud droplet activation, as well as evaporation of cloud droplet, though subject to the typical uncertainties inherent in the sectional approach' – please consider reword-ing this sentence to clarify what is meant by 'properties'.

We will reword "properties" as the shape of the distribution and number concentration of particles.

This is now included on page 6 lines 13-14.

6) Section 2: Could equations be added to describe the key microphysical processes?

We will add equations for the key processes and also provide some further details about their implementation.

More detailed description of the microphysical processes is found on pages 4-5 in the revised manuscript.

**7)** P5, L29: Please provide further details about the source for the coagulation kernels.

A reference for the source of the kernels will be added.

References have been added and the equations (3-5) are given on page 4 of the revised manuscript.

**8)** P5, L30-32: How is the dry size of the particle determined when the drizzle drop evaporates? Please clarify.

Upon evaporation of drizzle, the particle size is obtained by assuming that a single particle per droplet is released. The particle diameter is then the result from dividing the total aerosol mass with the bulk particle density. This will be stated more clearly in the text.

This is now explained in more detail on page 7, lines 31-33.

9) P8, L17: How do you define 'deeper and more massive shallow convection elements'?

By the deeper elements we refer to the cumulus clouds occasionally arising from about 400 m height (about the decoupling inversion height), supported by the build-up of heat and moisture from the surface. A more detailed description will be given in the manuscript.

This is noted on page 10, lines 26-31 in the revised manuscript.

**10)** P8, L32-34: In comparing the LEV3 and LEV4 simulations, it would be helpful to have a clearer description of the parameterization of drizzle formation/loss in LEV3 (the default UCLALES configuration). Perhaps this could be added earlier on in the model description.

We will add a more detailed description of these processes for the default UCLALES.

Description of the drizzle formation in the default UCLALES is given on page 8, lines 10-15 of the revised manuscript.

11) Fig 4: Where is LEV3 on panel 4b?

The LEV3 (as in the default UCLALES) does not contain a description for aerosols (apart from the prescribed CCN concentration used to yield the number of cloud droplets). Therefore this cannot be added. The panel 4b serves the purpose of illustrating the abilities of UCLALES-SALSA. We will make a better note of this in the manuscript to avoid confusion.

It is now explicitly stated on page 8, lines 8-9 that the UCLALES does not contain a description for aerosol and thus for scavenging. Moreover, it now says "Figure 4 shows the surface precipitation rate in LEV3 and LEV4 simulations as well as the rate of removal of sulphate aerosol embedded inside precipitating droplets in UCLALES-SALSA, ... " on page 12, lines 8-10.

**12)** P9, L33-35: How is scavenging treated in the below-cloud layers? Please consider adding this information.

Collision and collection processes are treated between all different particle and droplet classes using the coagulation equations. Upon collision between an aerosol particle and drizzle drop, the mass of the aerosol particle is moved to the drizzle bin in question. This will be elaborated on in the manuscript.

Scavenging by precipitation below-cloud is explained on page 7, lines 28-29 of the revised manuscript.

**13)** P10 L6: 'lack of representation for aerosol scavenging' – How is aerosol scavenging represented in LEV3? Consider adding this information earlier on in the text to help the reader in understanding these comparisons between the LEV3 and LEV4 simulations.

As stated in the response to comment No 11, LEV3 does not contain a representation for aerosols, and therefore the wet scavenging process is not represented by LEV3. This will be stated more clearly in the manuscript.

This is stated on page 8, lines 8-10.

**14)** P10, L11-12: 'LWP and rain water path show quite similar features as those obtained with a cloud system resolving model with interactive aerosols' – Please state these 'similar features' more explicitly.

The results in the low aerosol cases of this paper showed a similar depletion of cloud water caused by aerosol scavenging and drizzle. We will explain this more clearly.

It now says "Interestingly, the reduced cloud water and increased drizzle caused by depletion of aerosol, as shown by Figure 3 for LEV4, resemble the corresponding effects shown by (Yamaguchi and Feingold, 2015) ... " on page 12, lines 30-32.

**15)** Section 3.2: This section includes a detailed discussion of the simulation results for the case DYCOMS-II flight RF02, which was a marine stratocumulus case that took place off the coast of California. Would there be observations available that could be explicitly compared to the simulation output presented here?

We will add flight-mean estimates of LWP and surface precipitation to Figures 3 and 4, respectively. It is shown that the model results fit the observed values quite well, given the assumptions used in the model runs. A more detailed discussion will be added to the manuscript.

The observed estimates for LWP and surface precipitation retrieved from literature are now show in figures 3 and 4 and commented on page 11, lines 1-3 and lines 11-13.

**16)** P11, L22: Why was the drizzle formation switched off for this fog case?

Even though the drizzle formation was not explicitly used in these simulations, the fog droplets can grow freely, given the ambient conditions, and reach the size range when they begin to be removed by sedimentation. However, as stated also by Porson et al. (2011), the liquid water content remains relatively small, so explicit drizzle parameterization is not needed. We follow this notion to conform with their model setup.

This is noted on page 14, lines 23-25 in the revised manuscript.

**17)** P12, L19: Consider adding a table to describe the simulations A200, A400, A800 A400W.

We will add a table with the details of the simulation setups.

Table 2 has been added with the requested details and noted on page 15, line 10.

**18)** P13, L4-5: 'These findings illustrate the ability of the UCLALES-SALSA to provide a realistic description of not only the thermodynamic and microphysical properties. . ..' – Please consider if this statement would be better supported by explicitly showing model-observation comparisons in the manuscript.

Here, we refer to the results presented in Porson et al. (2011) and Price (2011). More elaborate discussion about comparing our model results with the afore mentioned data will be added to the manuscript.

This comment is considered in the added discussion on page 16, lines 10-30 in the revised manuscript.

**19)** P 13, L8-9 'growth rate is considerably lower than the observed'. . . 'see figure 5 in Porson et al., 2011' – are there observations that could be explicitly shown here to help the reader understand these comparisons?

We will use the data on fog layer growth presented in Porson et al. (2011) based on tethered balloon measurements. The data points are added to our Figure 9.

Observed data is now shown in Figure 10 (note the changed number!) in the revised manuscript.

**20)** P13, L21: 'These results point towards the importance of detailed representation of the microphysical processes.' This sentence does not appear to be finished – do you mean in cases of fog?

*Yes, in this context we refer to the fog case. We will reword the sentence.*

It now says "The results point towards the importance of detailed representation of the microphysical processes in cases of fog formation." on page 16, line 21.

**21)** P13, L22: 'UCLALES-SALSA does well' – Are you able to quantify what is meant by 'does well'?

This refers to the occurrence of the peak droplet number concentration mentioned in the next sentence. We will adjust the wording.

It now says "In particular, the size resolving microphysics in UCLALES-SALSA results in a peak number concentration in the fog droplet size distribution at approximately 25 μm in terms of the wet diameter, which agrees with the observed range between 20 μm and 25 μm based on the measurements presented in Price (2011)." on page 16, lines 22-24 in the revised manuscript

**22)** P13, L26 'UCLALES-SLASA also agrees well with observations' – again please quantify what is meant by 'agrees well' and consider showing model-observation comparisons in the manuscript.

Again we refer to Porson et al. (2011) where it is shown that droplet concentrations between 20 and 60 cm-3 were measured for this forg case. The droplet concentrations in the experiment A400 fit to this range, except when the fog layer eventually transforms into a shallow cloud later in the morning. We will discuss this in more detail in the manuscript.

Some new discussion is added on page 16, lines 26-31.

**23)** P13, L30: 'a more detailed land surface scheme is needed' – did you test any limiting cases?

No, we did not. The surface heat capacity was tuned to match the observed surface temperature, and the surface was assumed to be wet.

**24)** P14, L29-30 'very similar to the observations'. . . . 'even more resembles the observed properties' – Please consider showing these comparisons in the manuscript, likewise showing some model-observations comparisons would be helpful for understanding the model performance for the stratocumulus case.

Measured data is added to Figure 9 of the revised manuscript for fog layer growth (comment #19). Moreover, observed estimates of LWP and surface precipitation are now shown in Figures 3 and 4 (comment #15).

Changes are covered by earlier reponses.

**25)** P14, L29: If a realistic wind profile improved the model-measurement agreement – why was the case with winds not used as a default? Did you test A200W and A800W?

We wouldn't consider the no-wind simulations as the "default". Instead, they were considered first, because such a simple setup allows us to demonstrate the effect of aerosols specifically, which we are most interested in. With a realistic wind profile, the simulations were performed also with other aerosol concentrations. However, the mixing caused by wind shear dominates the growth rate of the fog layer over the initial aerosol concentration. Moreover, supersaturation inside the fog is also strongly affected by mixing, which makes the differences in fog droplet concentrations less clearly defined between the different aerosol concentrations.

Technical corrections:

1) P2,L10: Do you mean 'of' instead of 'off'? - *Corrected*

C42) P5, L20: 'Evolution of the drizzle droplet population' – should this read drizzle/rain since the upper diameter limit is 2mm? – *Done*.

3) Fig. 3a: Should HI be removed from the legend? - *Done*.

4) P12, L27: Do you mean Fig 8 as opposed to Fig. 9? - *No, Fig 9 is correct*.

5) P12, L31: Consider starting a new paragraph with the start of the Fig. 11 discussion. - *Done*.

6) P13, L13: Do you mean Fig. 9 as opposed to Fig. 10? There is no dashed line on

Fig. 9. - Corrected.

7) Fig. 1: What is the meaning of the light blue arrows on the dark blue for the drizzle

rain bins? What is the size range for the cloud droplets? <mark>– They were to represent the rain drop growth,</mark> <mark>but indeed might be misleading. They are removed. Size range is added.</mark>

8) Fig 2: Could g kg-1 be placed beside the color bar? - *Done*.

9) Fig 3: Could drizzle be added to the title of panel b? Also, please check legend for error in simulation names. - *Done*.

10) Fig 7: Please check units on the legend – did you mean m? - *Yes, corrected*.

All the corrections above have been implemented.

We thank Reviewer #2 for his/her comments and feedback. The largest changes in the manuscript in response to these comments will be the improved presentation of technical details in addition to the other more specific suggestions given by the Reviewer which will also be implemented in the revised manuscript. Below, we will list all of the Reviewer comments, followed by our response highlighted in *italics*.

The corrections to the manuscript regarding each comment are highlighted in italics.

**Paper content and a general impression**

\_

The submitted manuscript describes a newly-developed software package for research on aerosolcloud-precipitation interactions. The presented framework is composed of a modified (adapted) version of the free and open-source Large Eddy Simulation (LES) system UCLALES, coupled with a modified (extended) version of the SALSA aerosol process modelling package. The paper consists of brief description of both pre-existing software packages and of how they were adapted, extended and coupled to result in the UCLALES-SALSA system. Moreover, the capabilities of the developed tool, in particular its applicability to capture aerosol-cloud interactions, are exemplified for two different simulation set-ups.

The topic of the paper matches well with the current interests of the cloud-modelling community – the study fits into an active stream of development of modelling techniques to study aerosol-cloud-precipitation interactions in LES-type frameworks. These concurrent endeavours are not referenced comprehensively in the paper, though.

My main major concern is the misalignment of the paper content with the scope of GMD. The model description itself amounts to ca. 3 pages out of 15 (not counting figures or references), while the rest of the paper deals with the case studies. Of course, this is not the page-count that matters and the case studies do provide valuable examples and validation of the model capabilities. Yet, the model formulation and implementation – in my understanding the key elements of the GMD scope – are clearly described in not sufficient detail. There is not a single equation used in describing the new extensions to the UCLALES and SALSA, even though the authors admit that "coupling the extended SALSA module into UCLALES yields extensive changes in the thermodynamic core of the model". A key component newly introduced into the model formulation, representation of cloud droplet collisions and coalescence, is commented with just a single sentence without detailing the numerical method or its implementation. The reader is left without any information about software engineering aspects of the project (without studying the references, the reader would not even learn about the programming language in which the code is written; more importantly such aspects as parallelisation techniques, required environment and tools to use and extend the model are not mentioned and these cannot be guessed). There is no information on how the modified UCLALES and the extensions to SALSA are planned to be disseminated within the community, how existing UCLALES and SALSA users can benefit

from the described developments. If the authors would rather prefer to keep the paper focused on the case studies, and not the model formulation and implementation, I suggest submission of a revised version of the text to ACP or similarly scoped journal. The model code is also not publicly available as of now. It not only makes the paper not compliant with the GMD guidelines, but it also prevents me to fulfil the reviewer's duties – the GMD board clearly states that all papers "must be accompanied by the code, or means of accessing the code, for the purpose of peer-review", moreover the journal guidelines

"strongly encourage referees to compile the code, and run test cases supplied by the authors"1. The authors do not detail how the paper readers may reproduce the discussed results.

For the reasons listed above, I am requesting a second round of the review to follow. In my opinion, the manuscript requires substantial changes to reach a good level of readability and to match contemporary standards of research reproducibility pioneered by GMD. Nevertheless, let me repeat that the described research and the developed tool are of prime interest to the community. In particular, the described system is capable of simulating aerosol processing by clouds through activation-collision-deactivation cycles as well as resolving aerosol sources – features not widely available in other LES-type systems.

As a general statement, we will improve on the description of the model technical details in the manuscript. In the current version, it is stated that the model will be available through Github, and upon request before its release. We are currently still in the process of cleaning up the code for a public release and expect this to be finished shortly.

Lots of new details have been added especially to Section 2.1. Moreover, new Section 2.3 gives further information about the technical details.

**General remarks**

**Few references to other LES aerosol-cloud-precipitation interaction studies**

For the purpose of giving a comprehensive background, as well as of highlighting the unique features of UCLALES-SALSA, I strongly suggest supplementing the list of referenced works with some seminal

and/or recent papers on aerosol-cloud-precipitation interaction modelling with LES-type tools. In the list below, I suggest some that might be worth checking. The list is certainly not exhaustive, though:

- Lebo and Seinfeld, 2011: 10.5194/acp-11-12297-2011
- Ovchinnikov and Easter, 2010: 10.1029/2009D012816
- Andrejczuk et al. 2010: 10.1029/2010JD014248
- Shima et al. 2009 10.1002/qj.441
- Feingold et al. 1996: JGR 101 (D16)

The above references have been included in the revised manuscript on page 2, lines 10-15.

In particular, citing some of these works could support or otherwise require rewording of some statements:

• p. 2/line 9/10: "extensive simulations with more detailed and interactive ... schemes ... are relatively sparse"

The sentence "Nevertheless, some examples of such developments include the works of (Andrejczuk et al., 2010; ... " has been added on page 2, lines 14-15.

• p. 2/line 33: "innovative approach" (it is worth clearly stating precisely what is novel here) *This is stated more clearly, page 3, lines 7-9.*

• p. 4/lines 6-7: "not a computationally feasible approach"

It now says "However, although such two-dimensional frameworks have been developed (e.g. Lebo and Seinfeld, 2011), the approach is computationally highly demanding for large-eddy modelling applications spanning timescales of days while covering relatively large domains with high spatial resolution, which are pursued here" on page 6, lines 4-8.

We will adjust the manuscript as suggested by the Reviewer.

Lack of model formulation and implementation details

As outlined above, I strongly encourage the authors to increase the level of detail in which the model formulation and implementation is described. Here are some examples:

• p. 4/line 20: How the substepping is implemented? Are the grid-mean values kept constant for all substeps within a timestep (in particular, the supersaturation)?

Substepping is now described on page 5, lines 17-25 in the revised manuscript.

• p. 5/line 13,15: statements seem contrasting: "not knowing the wet droplet diameter exactly" but "bin mean cloud droplet wet diameter" is used, perhaps it is worth summarising clearly what are the variables and constants per bin for each spectrum, and which processes change them – a table would likely give best readability

We have reworded the first paragraph of Section 2.1.2. to avoid confusion. In addition, the sets of prognostic variables and their treatment within the aerosol, cloud droplet and precipitation sizebins is now explicitly mentioned on page 3, lines 27-28; page 6, line 10 and page 7, lines 18-19, respectively.

• p. 5/line 29-30: here the treatment of coalescence is described by just one single sentence without any reference. What are the numerics behind, how the kernels are supplied (if look-up tables, please detail interpolation method)?

Coagulation is now described more accurately on page 4, line 8 – page 5, line 7.

We will increase the level of technical details in the model description, including all the points above raised by the Reviewer. As also requested by Reviewer #1, equations describing the key model processes will be added.

**Simulation setup description**

The fact that two contrasting cloud regimes are simulated gives a nice opportunity to pick this a criterion for mentioning or not a given simulation parameter. I suggest thus creating a table listing all model parameters that needed to be changed (or where arbitrarily switched on or off) in order to make the simulation depict fog instead of stratocumulus. This could perhaps allow to shorten a bit the setup descriptions in the text, the initial profile given by eq. 1-4 could then be part of the table (why not just cite the relevant equations in the paper in which the DYCOMS profiles where defined). If adaptive timestepping was used, please provide some statistics on the timestep values for the two different setups. If a spinup period is used for model initialisation, please clearly indicate which processes are on or off for how long, and what are other differences between the spinup and the rest of the simulation.

We will include more detailed information about the model setups as well as the spinup configuration. We will also elaborate on how the setup for fog simulations differs from the stratocumulus case. These differences are mainly comprised of model resolution, surface conditions and the input sounding. Adaptive timestep is used – we will add more detailed information about the values to the manuscript.

The description of the fog experiments is revised and now explicitly mentions the differences in the configuration with respect to the stratocumulus case, i.e. on page 14, lines 7-15 and lines 22-23. In addition, the case specific settings for the fog simulations are now divided into a new Section 4.1.1 (regarding the initial aerosol concentration and wind conditions). The spinup is detailed for the stratocumulus case on page 9, lines 20-23, and for the fog case on page 15, lines 1-2. Adaptive timestep is detailed for stratocumulus on page 9, lines 25-26 and the fog case on page 14, line 12.

Also, some of the model features advertised in the beginning of the paper seem not used in the simulations (e.g., condensation of precursor gases and new particle formation mentioned on p. 3/line 30) – please state it explicitly. In contrast, features such as inclusion of the diurnal cycle or the soil

energy balance are not mentioned in model description part.

The condensation of aerosol precursors was active in the model simulations, but it's effect was negligible in the current simulations setups. New particle formation was not active, although available. We will mention these and the other features suggested by the Reviewer.

Condensation of aerosol precursors is now mentioned in the model description, page 5, lines 9-10 and new particle formation on lines 27-29. Soil energy balance is given in the case descriptions since its different for stratocumulus and fogs: page 9, line 19 and page 14, lines 15-22. Diurnal cycle is mentioned in the model description on page 8, lines 24-25.

Statements such as "large cloud droplet are considered as drizzle" (p. 8/line 27) or "the surface heat capacity is used as a tunable parameter" (p. 12/line 5) call for numbers.

We will add this information in the manuscript.

Drizzle threshold diameter is mentioned (now on page 11, line 8), Heat capacity is further explained on page 14, lines 20-22

If I understood correctly, presented simulations lack aerosol sources. In contrast, the setups like DYCOMS-II implicitly assume an infinite reservoir of CCN brought in to the domain by advection. If that is correct, this difference is worth mentioning and perhaps discussing.

This is true, as well as the fact that the interpretation of the model simulations indeed does change when switching from the default UCLALES with prescribed CCN concentration to UCLALES-SALSA with a more dynamic description. The latter points more towards a "Lagrangian" simulation in the sense that the depletion of aerosols by clouds and precipitation resembles a domain moving with the flow. We will add discussion of this in the manuscript.

This is added on page 12, lines 22-26.

**Aerosol processing nomenclature**

Depending on the community "aerosol processing" is associated with different processes if put out of context. Please clearly state, at least in the abstract and introduction, whether chemical processing or collisional processing is addressed. Especially, since condensation of sulphates is mentioned on page 3.

Despite the condensation of aerosol precursors included in the model, for now the model does not have chemical processing. This will be stated more clearly.

This is mentioned on page 7, line 4.

**Section scope**

The introduction section mentions such, distant from the scope of the paper, matters as challenges in climate modelling, arctic temperatures changes, decrease in fog occurrence in Central Europe. For fellow cloud modellers, the links between those topics and the paper scope might be "obvious", for other members of the GMD audience these will seem puzzling, tough. Please either elaborate on how and why these topics are related with the development of UCLALES-SALSA or keep the introduction closer to the paper scope.

Since this modelling work ultimately aims at providing a research tool to improve the above mentioned features in global and regional climate models, we will keep these topics in the introduction, but will provide more in depth description, as suggested by the Reviewer.

We have clarified the motivation for this discussion on page 2, lines 16-27 and page 3, lines 2-6.

The DYCOMS-II section uses up to three-digit section numbering (e.g., 3.2.1) while the fog case is just divided between two case description and Results subsections. I suggest some work on restructuring the two sections to be more similar in both section numbering and, more importantly, the level of detail.

We will break Section 4 into smaller divisions.

The fog case experiment details are now divided into a new Section 4.1.1

The specific suggestions below will be implemented into the manuscript unless otherwise mentioned.

Specific comments and rewording/correction suggestions

**• Paper-wide:**

- drizzle  $\rightarrow$  precipitation (in particular in the title, the model is not limited to drizzle and since one of the quantities analysed is the surface precipitation rate, the simulated precipitation is by definition not drizzle) *Done*

– aerosols → aerosol (e.g., in the title, I don't have a strong opinion on it - just a suggestion)  $\frac{Done}{Done}$

- computational burden  $\rightarrow$  computational cost Done

- high computational burden  $\rightarrow$  resource intensive, etc *Computational cost is now used*

– interactive, fully interactive scheme, interactive description of particles – please explain what you mean exactly (by explaining which models are non-interactive), especially as it is mentioned in the title *The terminology has been made more clear and the title is justified by that UCLALES-SALSA tries to capture the aerosol-cloud coupling explicitly in both directions as well as their impacts on the dynamics. This is stated more clearly on page 3, lines 7-9.*

– please ensure that acronyms are explained on first occurrence (e.g., SALSA is only deciphered in section 2.1) This is the first occurrence in the text.

**• Abstract:**

− line 1: impacts of  $\rightarrow$  impacts on **Done**

– line 1,3: improved over what?, more sophisticated than what?, what kind of observations?
 Please be precise, please try to cater to a wider community, please make sure that the abstract summarises the presented research – global climate and gaps in observations seem not relevant enough to pop up in the very first sentences of the abstract *We have reworded the abstract to avoid confusion. Global modelling and lack of observations are the most essential key motivation for building the UCLALES-SALSA in the first place so mentioning them in couple of sentences of the abstract is justified.*

- line 4: model, coupled  $\rightarrow$  model (LES), coupled **Done**

– line 5/6: "microphysical model components" is vague – please state if you refer to SALSA or something else as well Done

– line 6: "strategies for … bin layouts" reads awkward, perhaps the keyword discretisation could help to better convey what is meant? I understand bin layout as a parameter of a given simulation, what is perhaps worth mentioning in the abstract is how the modelled particles are classified and which classes of particles are subject to which processes *The word discretization is now used. Going into implementation and process-level details is too much for abstract – these are described thoroughly in the text.*

– line 8: "computational cost of the model acceptable" – this is not only subjective but also likely to be objectively false soon *It now says "as low as possible"*

− line 8: two different cases: one comprising a case with marine stratocumulus . . . → two different simulation setups: the DYCOMS-II marine stratocumulus setup  $\frac{Done}{Done}$

– line 9-10: It is shown that, in both cases, . . . Done

– line 13: In radiation fogs, the growth *Done*

− line 14: strongly affects  $\rightarrow$  strongly affect Done

**• Introduction:**

– p. 1/line 18: simulators → simulations  $\frac{Done}{D}$

– p. 1/line 19: Moeng 1984 – please either use a few references to support the use of LES

for decades or cite a recent review paper (preferably) *References added, page 1, line 21.*

– p. 2/line 7: please mention also particle-based models (in addition to bulk, modal and sectional) *Mentioned, page 2, line 11.*

– p. 2/line 10: "mostly due to their high computational burden" – isn't it the multi-scale and multidisciplinary nature of clouds that limits us most and not the computer power? *Yes, but building models much more elaborate than what we can afford to use (at least widely) is a fully relevant consideration in the context of this sentence.*

– p. 2/line 25: "fogs also feature many different aspects" – please reword <mark>It now says "Although in many ways driven by the same principles as clouds, fogs also feature many unique aspects considering their evolution." on page 2, lines 32-33.</mark>

– p. 2/line 30: four references given to support statement that fogs are affected by anthropogenic emissions – please try to keep balance with the paper and journal scope *This issue is not straightforward and the cited references may give the necessary background information for some readers.*

– p. 3/line 6,7: "well-characterised" vs. "findings of" could hint that one is superior to the other, please reword *We have removed "well-characterized" to balance out*.

– p. 3/line "previous model versions" versioning suggests something linear, in this context we are rather faced with multiple diverging branches, please try to avoid the version when giving a precise version number would be tricky *Reworded*, *page 3*, *line 17-18*.

**• Section 2:**

– p. 3/line 16: drizzle/rain → precipitation Done

– p. 3/line 17: since the number 10 is just a setting for a particular simulation, perhaps it is worth explaining instead how the bins are laid out (logarithmically?) *Done, page 4, line 3-4.*

– p. 3/line 18: what does the Bergman et al. citation refer to? (10 bins?, Figure?) The discretization with 10 bins, reference moved to point to this more clearly.

– p. 4/line 8: "parallel bins" is hard to understand *This is explained more clearly, page 6, lines 10-15.*

– p. 4/line 15: non-chronological order of citations *Corrected*.

– p. 4/line 16: "very fast relative changes" is vague, isn't it anyhow the stiffness of the governing equations due to presence of multiple size scales that is the crux of the matter? *The subsetpping issues are described in more detail now on page 5, lines 17-25.*

– p. 4/line 32/33: "unwanted discontinuities in … calculation", please reword so it is clear what is discontinuous *Reworded, page 6, lines 29-30*.

– p. 5/line 8: I suggest removing the last sentence of this paragraph *Done*

– p. 5/line 9: drizzle → precipitation Done

– p. 5/line 12: please rephrase, perhaps referring to statistical moments resolved within each bin would make it clearer *Reworded*, *page 7*, *line 6-8*.

– p. 5/line 21: please rephrase so that it is clear that wet diameter is the relevant quantity for condensation and collision, currently the sentence suggests that drizzle condensational growth is critical to produce realistic precipitation **Done**, *page 7*, *lines 16-17*.

– p. 5/line 22: does SALSA share the implementation of the Abdul-Razzak and Ghan (2002) scheme with some other (open-source?) model? *It does not*.

– p. 5/line 28: please explain (mathematically) how the bin layout is formulated *This is explained in more detail on page 7, lines 24-27.*

– p. 5/lines 31-32: please refer to the "aerosol processing" in the sentence **Done**, *page 7*, *lines 32-3*.

– p. 5/lines 31-32: perhaps citing Mitra et al. 1992 could be used to support the assumption? *See above*.

– p. 6/line 10: "default version" might mean something different for each user, please be specific *We now refer to this simply by "UCLALES"*.

– p. 6/line 20: "raising the number of prognostic scalars" → "increasing the number of advected scalars from O(?) to O(100) *Done, page 9, line 1.*

– p. 6/line 22: please hint the level of concurrency used – otherwise just the computer type makes the statement very vague *Done, page 9, lines 1-5.*

**• Section 3:**

– p. 6/line 31: is there any limit on the magnitude of the supersaturation during the initialisation? *Currently no.*

– p. 7/line 22: would one of "Reference case"/"Reference setup"/"Reference run" be more apt than "Default case"? *"Reference case" is now used.*

– p. 7/line 27: from previous statements, I understood model initialisation to be equivalent the spinup period, here it seems to mean pre-time-zero calculation – please use consistent wording *It now says "model startup"*.

– p. 7/line 28: please reserve the word "parallel" for calculation concurrency

– p. 7/line 28: please define somewhere the "default UCLALES configuration" – again, this

might mean different settings to different users *It now says "UCLALES with bulk microphysics"*.

– p. 7/line 29: "default UCLALES" – does it refer to the "default case", "default configuration" or something else *Referred again simply as UCLALES*.

– p. 8/line 17: the domain-mean plot discussed was likely not the basis for statements"massive shallow convective cumulus elements" or open cells; please clearly define where

you discuss the figure This is now explained more clearly on page 10, lines 26-30.

– p. 9/line 20: "By the same token" sounds strange to me (but I'm not a native speaker) Switched to "argument".

– p. 9/line 29-31: please reword, "performs this task with very high detail" could be omitted, the use of "beyond" is unclear here *Reworded, page 12, lines 10-12.*

– p. 9/line 32-33: another example where the reader can be puzzled about differencesspinup and model initialisation *Reworded, page 12, lines 12-13.*

– p. 10/line 12: please reword "model with interactive aerosols" *Reworded, page 12, lines 29-32.*

– p. 10/line 24: "to the their" → "to their" **Done**

– p. 10/line 33: I assume this means very small in one simulation and non-existent in the other, please reword **Reworded**, *page 13*, *lines 19-20*.

– p. 11/line 5: isn't coagulation part of the processing **Reworded**, *page 13*, *lines 24-27*.

• Section 4:

– p. 11/line 28: "water surface pressure" → saturation vapour pressure? It now says equilibrium saturation ratio, page 14, line 32.

– p. 12/line 7-8: "is not available" suggests lack of availability, here it was simply not partthe setup <mark>No,</mark> the information was not available.

– p. 13/line 5: "-radiative" → "-radiation" Done

– p. 13/line 5: if "and feedbacks" is needed, please explain how do you differentiate them from interactions *Done, page 16, lines 4-5*.

– p. 13/line 29: "connect the aerosol concentration into fog existence" – I suggest rewording *It now* says "couples with fog occurence" on page 16, line 33-34.

**• Conclusions:**

– p. 13/line 33: "A new large-eddy simulation model" suggests some new fluid dynamics

methodology, while the novelty is elsewhere – please reword *Reworded*, *page 17*, *line 2*.

– p. 14/line 5: please precise what type of processing (i.e., non-aqueous-chemistry related) Reworded, page 17, line 7.

– p. 14/line 27: "observed behavior Price" needs a parenthesis Done

- References:
- line 23: korolev → Korolev <mark>Done</mark>
- line 28: kokkola → Kokkola <mark>Done</mark>
- line 30: korhonen → Korhonen <mark>Done</mark>

• Figure 2: please sort out the background colour issue in panel b There is in fact no background color issue. In UCLALES-SALSA the condensed water in aerosol particles is classified as "cloud water" which, even though small, becomes visible because of the log-scale of the colormap. We will note this in the caption.

**Introducing** UCLALES-SALSA v1.0: a large-eddy model with interactive sectional microphysics for aerosolsaerosol, clouds and drizzleprecipitation**

Juha Tonttila1,2, Zubair Maalick4, Tomi Raatikainen3, Harri Kokkola2, Thomas Kühn4, and Sami Romakkaniemi2

1Karlsruhe Institute of Technology, Karlsruhe, Germany

[revised manuscript text omitted]